# The impact of reproductive factors on the metabolic profile of females from menarche to menopause

Gemma L. Clayton [1,2,4] ✉, Maria Carolina Borges[1,2,4] & Deborah A. Lawlor [1,2,3]

We explore the relation between age at menarche, parity and age at natural menopause with 249 metabolic traits in over 65,000 UK Biobank women using multivariable regression, Mendelian randomization and negative control (parity only). Older age of menarche is related to a less atherogenic metabolic profile in multivariable regression and Mendelian randomization, which is largely attenuated when accounting for adult body mass index. In multivariable regression, higher parity relates to more particles and lipids in VLDL, which are not observed in male negative controls. In multivariable regression and Mendelian randomization, older age at natural menopause is related to lower concentrations of inflammation markers, but we observe inconsistent results for LDL-related traits due to chronological age-specific effects. For example, older age at menopause is related to lower LDL-cholesterol in younger women but slightly higher in older women. Our findings support a role of reproductive traits on later life metabolic profile and provide insights into identifying novel markers for the prevention of adverse cardiometabolic outcomes in women.

Markers of women's reproductive health, such as age at menarche, parity and age at menopause, have been associated with several common chronic conditions, including cardiometabolic diseases[1–6] and breast, ovarian and endometrial cancer[7–12]. Some attempts have been made to explore the extent to which these associations are causal, as opposed to explained by residual confounding, using approaches such as Mendelian randomisation (MR) and negative control designs, which are less prone to bias by key confounders from conventional observational studies. MR studies suggest a direct positive effect of age at menarche on breast cancer and an indirect inverse effect via body mass index (BMI)[13], as well as a possible bidirectional relationship between age at menarche and BMI[13,14]. MR also supports a protective effect of older age at first birth on type 2 diabetes and cardiovascular diseases[15] and lower mean levels of BMI, fasting insulin and triglycerides in women and men[16], while a partner negative control study provides some evidence of a 'J-shaped' effect of parity on

coronary heart disease risk[5]. In addition, evidence from MR studies indicate that older age at menopause increases the risk of breast, endometrial and ovarian cancer, reduces the risk of bone fractures and type 2 diabetes, and do not substantially affect BMI or cardiovascular diseases risk[17].

Metabolites could act as mediators of the relationship of reproductive markers, and related hormonal changes, with chronic diseases[18–20]. Determining the effect of women's reproductive markers on multiple metabolites would be the first step to exploring this and could provide crucial insights into mechanisms underlying women's long-term health. We have previously shown marked changes in metabolites, such as lipids, fatty acids, amino acids and inflammatory markers during pregnancy[20], through the menopausal transition[21], and among women on hormonal contraceptives containing oestrogen[22]. Many of these same metabolic measures are also related to cardiovascular diseases[19] and some cancers[23–26]. The aim of this paper is to

[1]MRC Integrative Epidemiology Unit, University of Bristol, Bristol, UK. [2]Population Health Sciences, Bristol Medical School, University of Bristol, Bristol, UK. [3]NIHR Bristol Biomedical Research Centre, Bristol, UK. [4]These authors contributed equally: Gemma L. Clayton, Maria Carolina Borges. ✉e-mail: gemma.clayton@bristol.ac.uk

explore the extent to which women's reproductive markers have a causal effect on 249 metabolic measures (covering lipids, fatty acids, amino acids, glycolysis, ketone bodies and inflammatory markers). We focus on three reproductive traits that represent key events in women's reproductive lives: (i) age at menarche, a marker of puberty timing, (ii) parity, a marker of repeated exposure to the physiological challenges of pregnancy, and (iii) age at menopause, a marker of reproductive aging. We explore the causal relationships between these reproductive markers and metabolic measures by triangulating evidence[27] across multivariable regression, a negative control design (for parity only), and MR (Fig. 1). Given each of these approaches has unique strengths and limitations, results that agree across them are less likely to be spurious[27].

In this paper we show that reproductive factors are likely to impact females' metabolic profile later in life. Evidence supporting a relation between later pubertal timing and a less atherogenic metabolic profile is largely explained by adult BMI. Findings linking higher parity to a more atherogenic profile were supported by the negative control analyses but imprecisely estimated in Mendelian randomisation. Evidence supporting a relation between slower reproductive aging and a less atherogenic metabolic profile was mostly observed among younger women. These results could contribute to identifying novel markers for the prevention of adverse cardiometabolic outcomes in women and/or methods for accurate risk prediction.

## Results

We used data from 65,699 UK Biobank female participants with 249 metabolic measures quantified by nuclear magnetic resonance (NMR). Self-reported age at menarche (in years), parity (in number of live born children) and age at menopause (in years) were reported at baseline when participants mean age was 56 years old (range: 38–73). NMR metabolites were measured on blood samples taken at baseline or first repeat assessment (more details in methods). The characteristics of these participants are shown in Table 1 (and split by each of our reproductive markers (categorised) in Supplementary data 1–3). At

recruitment (baseline) women were aged (mean) 56 (SD = 8.0) years, 21% drank three or four times a week and 40% were previous/current smokers. 81% of women had one or more live births whilst the mean age of menarche was 13 years (SD = 1.3). 59% (37,248) women reported they went through a natural menopause with a mean age of menopause of 49.7 years (SD = 5.1). Supplementary data 4 shows the distribution of NMR metabolic measures among UK Biobank females. The proportion of women with missing data across metabolic measures ranged from 0.3% to 6.1%.

We used three approaches relying on different assumptions to explore the causal role of women's reproductive markers on later life metabolic profile. For the first approach ('multivariable regression'), we used linear regression models to estimate the association of reproductive markers with metabolic measures after adjusting for age at baseline, education and body composition at age 10. In sensitivity analyses, for the 55 non-derived metabolites, we categorised age at menarche, parity and age at natural menopause, tested for a linear trend and, where there was evidence of non-linearity, fit restricted cubic splines. For the second approach ('negative control design' – only applicable for parity), we used linear regression models to test whether number of live born children in men was associated with their metabolic measures. Men do not experience the repeated physiological stress of pregnancy but are likely to demonstrate the same associations of confounders (eg. socioeconomic position, BMI, smoking) with number of live births. Therefore, similar associations of number of live births with metabolic measures between men and women would indicate bias (e.g. due to confounding) rather than a causal effect of being exposed to the physiological stress of pregnancy on women's metabolic profile. For the third approach ('MR'), we selected single nucleotide polymorphisms (SNPs) as genetic instruments for each reproductive marker from previous genome-wide association studies (GWAS) and performed two-sample MR to estimate the effect of reproductive markers on metabolic measures using the standard inverse variance weighted (IVW) method. For both multivariable regression and MR analyses, we adopted P-value < 0.00093,

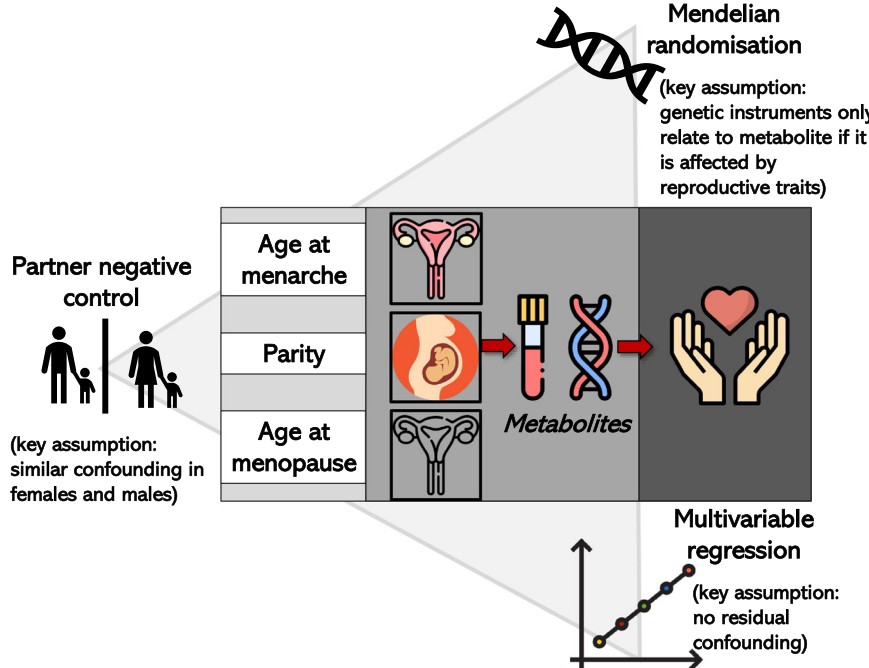

**Fig. 1 | Infographic summarising the different approaches taken to assess the relationship between reproductive traits and metabolites.** The figure illustrates key assumptions and sources of bias for each method (and differences across methods) in the context of our study. An exhaustive review of assumptions/biases for each method is outside the scope of this work. However, we acknowledge that there are other potential sources of biases that could affect findings such as selection bias related to the low response in UK Biobank.

## Table 1 | Distribution of characteristics of UK Biobank participants (females only) with NMR metabolomics data

| N = 65,699 | |
|---|---|
| Age when attended assessment centre, mean (sd) | 56.4 (8.0) |
| Age in 5-y groups, n (%) | |
| 40- | 6623 (10.1) |
| 45- | 8781 (13.4) |
| 50- | 10,378 (15.8) |
| 55- | 12,299 (18.7) |
| 60- | 16,089 (24.5) |
| 65- | 11,529 (17.5) |
| Ethnic background, n (%) | |
| White | 62,063 (94.8) |
| Mixed | 420 (0.6) |
| Asian | 1069 (1.6) |
| Black | 1059 (1.6) |
| Chinese | 220 (0.3) |
| Others | 609 (0.9) |
| Qualifications, n (%) | |
| None of the above | 10,955 (16.9) |
| O level or CSEs or other | 25,619 (39.4) |
| A level | 7626 (11.7) |
| College/ university | 20,749 (31.9) |
| Townsend deprivation index at recruitment, mean (sd) | −1.4 (3.0) |
| Comparative body size at age 10, n (%) | |
| Average | 32,767 (50.7) |
| Thinner | 20,416 (31.6) |
| Plumper | 11,435 (17.7) |
| Body mass index (BMI), mean (sd) | 27.1 (5.2) |
| Alcohol intake frequency., n (%) | |
| Daily or almost daily | 10,516 (16.0) |
| Three or four times a week | 13,712 (20.9) |
| Once or twice a week | 16,872 (25.7) |
| One to three times a month | 8655 (13.2) |
| Special occasions only | 9630 (14.7) |
| Never | 6186 (9.4) |
| Smoking status, n (%) | |
| Never | 38,915 (59.5) |
| Previous | 20,609 (31.5) |
| Current | 5834 (8.9) |
| Age when periods started (menarche), mean (sd) | 13.0 (1.6) |
| Categories of age at menarche, n (%) | |
| <13 years | 24,811 (39.0) |
| 13–14 years | 28,057 (44.1) |
| >14 | 10,762 (16.9) |
| Number of live births, n (%) | |
| 0 | 12,367 (18.9) |
| 1 | 8900 (13.6) |
| 2 | 28,526 (43.5) |
| 3+ | 15,767 (24.0) |
| Had menopause, n (%) | |
| Premenopausal | 15,418 (24.5) |
| Postmenopausal | 37,248 (59.1) |
| Surgical | 7564 (12.0) |
| Other | 2758 (4.4) |

## Table 1 (continued) | Distribution of characteristics of UK Biobank participants (females only) with NMR metabolomics data

| N = 65,699 | |
|---|---|
| Age at menopause (last menstrual period), mean (sd) | 49.7 (5.1) |
| Categories of age at menopause, n (%) | |
| <40 | 1520 (4.1) |
| 40–44 | 3378 (9.1) |
| 45–49 | 8903 (23.9) |
| 50–51 | 8740 (23.4) |
| 52–54 | 9144 (24.5) |
| 55+ | 5606 (15.0) |
| Statin use - Nurses int., n (%) | |
| No Statins | 58,279 (88.7) |
| Statins | 7420 (11.3) |
| Ever used hormone-replacement therapy (HRT), n (%) | |
| No | 40,330 (61.7) |
| Yes | 25,036 (38.3) |
| Years on HRT, n (%) | |
| Never | 40,330 (64.8) |
| 0–2 | 6294 (10.1) |
| 3–6 | 5504 (8.8) |
| 7–10 | 5192 (8.3) |
| >10 | 4908 (7.9) |

Age at natural menopause therefore excluded women who had not yet gone through the menopause or who had a surgical menopause or who answered 'other'.

which accounts for the approximate number of independent tests as detailed in 'Statistical analyses'.

### Age at menarche

In the main multivariable regression analyses (adjusting for age at baseline, education and body composition at age 10), older age at menarche was associated with higher concentrations of glutamine, glycine, albumin, apolipoprotein A1, cholines, phosphatidylcholines, and sphingomyelins but lower concentrations of alanine, branched-chain amino acids (isoleucine, leucine and valine), aromatic amino acids (phenylalanine and tyrosine), fatty acids (monounsaturated fatty acids (MUFA), omega-3 polyunsaturated fatty acids (PUFA), and saturated fatty acids (SFA)), glycolysis-related metabolites (glucose, lactate, pyruvate), acetoacetate, and glycoprotein acetyls (GlycA) ($P < 0.00093$) (Fig. 2 and Supplementary data 5). Older age at menarche was also associated with numerous lipoprotein-related traits at $P < 0.00093$, particularly with higher numbers of particles, size, and lipid content in high-density lipoproteins (HDL) and lower numbers of particles, size, and lipid content in very low-density lipoproteins (VLDL) (Fig. 2). The associations of age at menarche with HDL-related traits were mostly due to larger HDL subclasses (i.e. medium, large and very large particles), while associations with VLDL-related traits were observed across VLDL subclasses (Supplementary Fig. 1 and Supplementary data 5). In sensitivity analyses with further adjustments for BMI, smoking and alcohol status at baseline, findings for an association of older age at menarche were largely or completely attenuated towards the null for most metabolic measures with few exceptions, such as glutamine, glycine, omega-3 PUFA, pyruvate, lactate, and acetoacetate (Supplementary Fig. 2). There was evidence of non-linearity between categories of age at menarche (<13, 13–14, >14 years) and 17 metabolites (Supplementary data 6 and Supplementary Fig. 3). Restricted cubic spline models (with 3 knots at ages 11, 13, and 15 years)

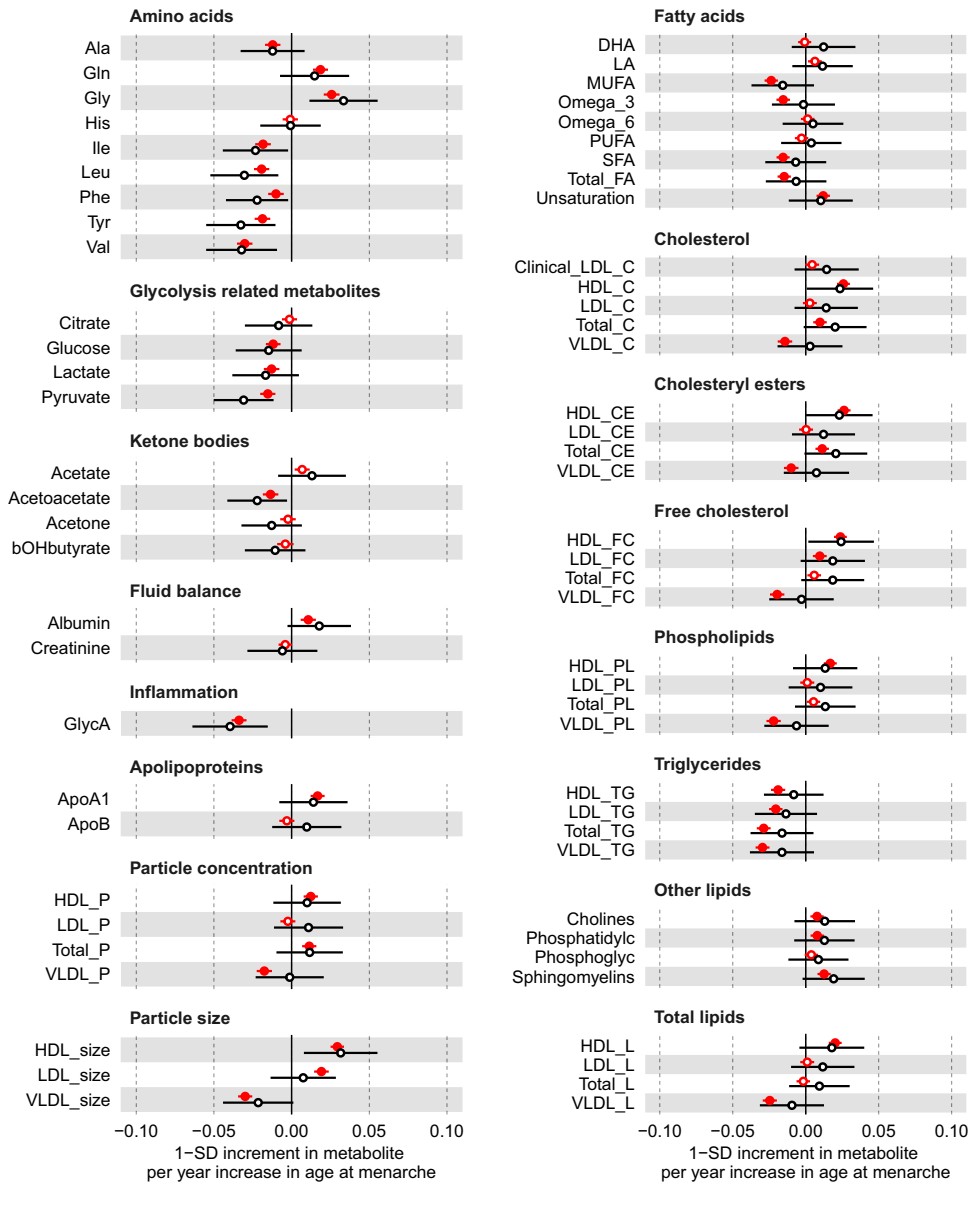

**Fig. 2 | Multivariable regression (red) and Mendelian randomisation (black) estimates for the associations between older age at menarche and metabolic measures.** Results are mean differences presented as standard unit changes in metabolic measure per 1 year increase in age at menarche. Circles denote the mean differences and indicate *p*-value < 0.00093 (filled circles) or ≥0.00093 (hollow circles). Horizontal bars denote 95% confidence intervals. Multivariable regression models (ordinary least squares, two-sided regression coefficients reported) were adjusted for age at recruitment, body size at age 10 and education (*N* = 61,920). Mendelian randomisation models were estimated using the inverse variance weighted method (*N* = 62,209). Abbreviations for Figs. 2–4 and 6 are given in 'Abbreviations' at the end of the manuscript. Source data are provided as a Source Data file.

generally showed an increase in albumin, apolipoprotein A1, cholines, docosahexaenoic acid (DHA), linoleic acid (LA), and phosphatidylcholines with older age at menarche until approximately age 13, in line with our linear association, and then began to flatten and/or decrease (Supplementary data 7 and Supplementary Fig. 4). Whilst older age at menarche was related to a decrease in GlycA until ~13 years and then began to flatten.

For the MR analyses, we selected 389 SNPs as instruments for age at menarche, which explained 7.4% of its phenotypic variance with a corresponding mean F statistic of 63 (Supplementary data 8). Overall, MR estimates using IVW were in agreement with multivariable regression estimates in direction and magnitude (Fig. 2 and Supplementary Fig. 1); however, due to the higher degree of uncertainty for IVW estimates, no result passed our threshold for multiple testing

correction (*P* < 0.00093). Following reviewer's comments, we repeated the IVW analyses for a larger sample of women (*N* = 216,514−239,803) for the eight biomarkers assayed using clinical chemistry techniques that matched measures in the NMR metabolomics platform − i.e. albumin, apolipoprotein A1, apolipoprotein B, glucose, HDL-cholesterol, low-density lipoprotein (LDL)-cholesterol, total cholesterol, and triglycerides. These results provided further evidence of older age at menarche being related to higher albumin, apolipoprotein A1, HDL-cholesterol, and lower triglycerides (*P* < 0.00093) (Fig. 5). Given the a priori evidence of bidirectional effects between age at menarche and BMI, we also performed multivariable IVW accounting for adult BMI to estimate the direct effects of age at menarche on metabolic measures, which resulted in estimates partly or completely attenuated to the null for most metabolic

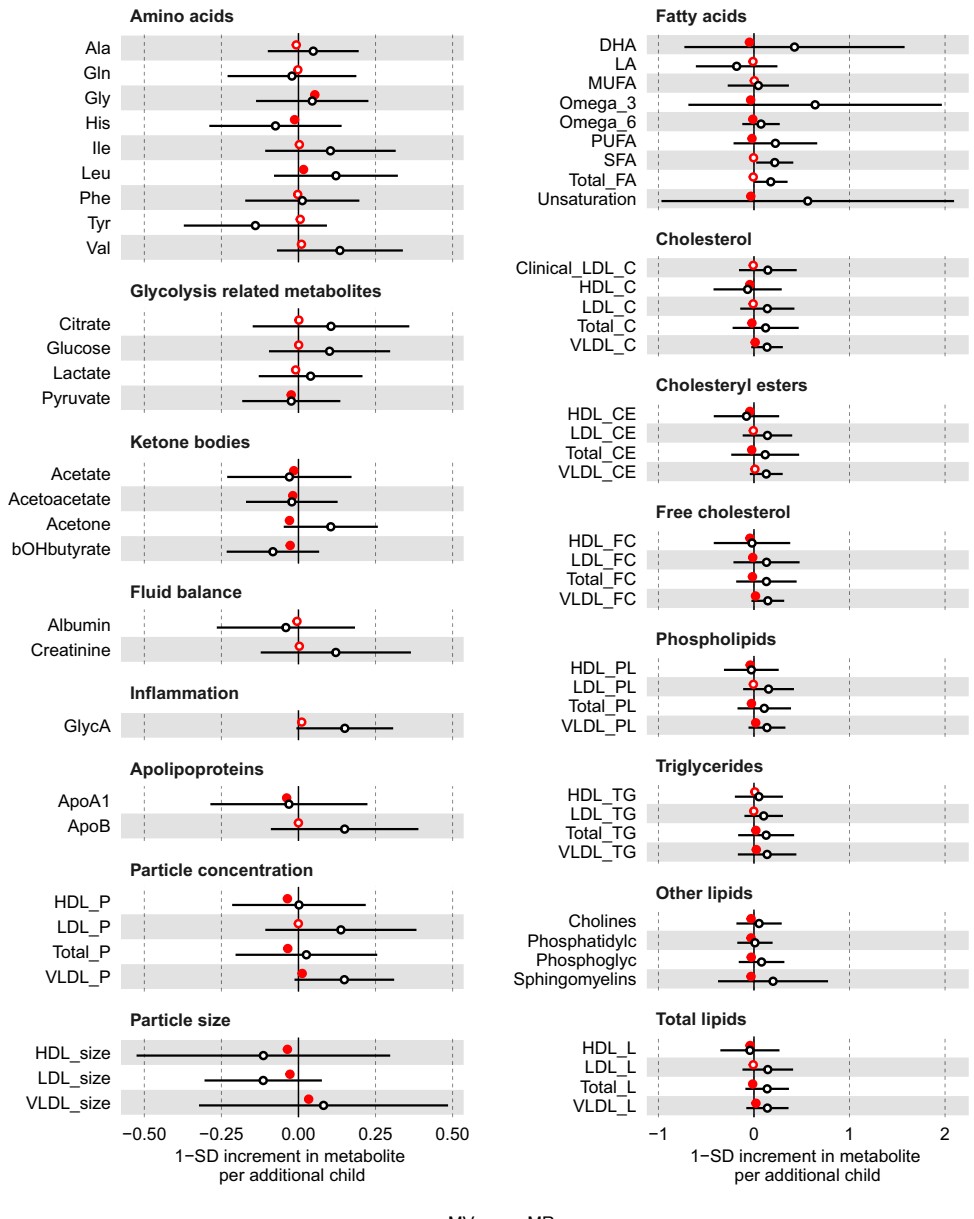

**Fig. 3 | Multivariable regression (red) and Mendelian randomisation (black) estimates for the associations between higher parity and metabolic measures.** Results are mean differences presented as standard unit changes in metabolic measure per 1 additional birth. Circles denote the mean differences and indicate p-value < 0.00093 (filled circles) or ≥0.00093 (hollow circles). Horizontal bars denote 95% confidence intervals. Multivariable regression models (ordinary least squares, two-sided regression coefficients reported) were adjusted for age at recruitment, body size at age 10 and education ($N$ = 63,652). Mendelian randomisation models were estimated using the inverse variance weighted method ($N$ = 62,209). Abbreviations for Figs. 2–4 and 6 are given in 'Abbreviations' at the end of the manuscript. Source data are provided as a Source Data file.

measures with few exceptions, such as glutamine and glycine (Supplementary Figs. 5 and 6).

**Parity**

In the main multivariable regression analyses (adjusting for age at baseline, education and body composition at age 10), higher parity was related to higher concentrations of glycine and leucine, but lower concentrations of histidine, fatty acids (DHA, Omega 3, Omega 6 PUFA), pyruvate, ketone bodies (acetate, acetoacetate, acetone and β-hydroxybutyrate), and apolipoprotein A1 ($P$ < 0.00093) (Fig. 3 and Supplementary data 9). Higher parity was also associated with numerous lipoprotein-related measures at $P$ < 0.00093, particularly with lower and higher number of particles, size, and lipid content for HDL and VLDL, respectively, as well as lower size of LDL particles

(Fig. 3). The associations of parity with lipoprotein-related measures were observed across most VLDL and HDL subclasses, whereas associations with LDL-related measures were mostly driven by larger LDL particles (Supplementary Fig. 7 and Supplementary data 9). In sensitivity analyses with further adjustments for BMI, smoking and alcohol status at baseline, higher parity associations were consistent for glycine, histidine, fatty acids, pyruvate, ketone bodies, apolipoprotein A1, and partly attenuated towards the null for VLDL- and HDL-related traits (Supplementary Fig. 8). There was some evidence of non-linearity between parity (0,1,2,3+) and 28 metabolites (Supplementary data 6 and Supplementary Fig. 9). However, restricted cubic spline models (with knots at 1, 2, and 3) generally showed monotonic relationships for those with no to four pregnancies, consistent with the main analysis models (Supplementary data 10 and Supplementary Fig. 10).

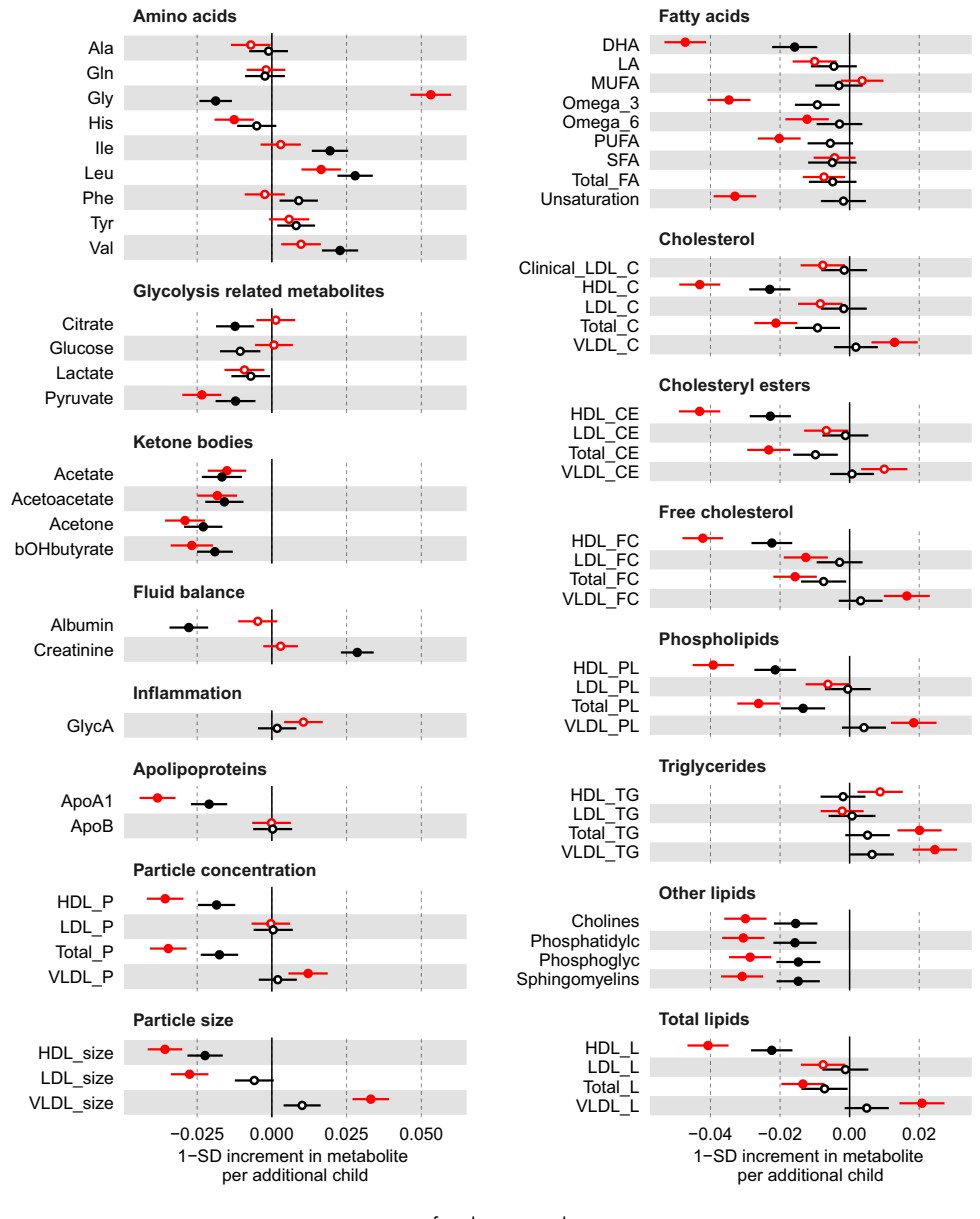

**Fig. 4 | Multivariable regression estimates for the associations of parity (females, red) or number of children (males, black) with metabolic measures: negative control analyses Models adjusted for age at baseline, education, and body composition at age 10.** Results are mean differences presented as standard unit changes in metabolic measure per 1 year increase in age at menopause. Circles denote the mean differences and indicate *p*-value < 0.00093 (filled circles) or ≥0.00093 (hollow circles). Multivariable regression models (ordinary least squares, two-sided regression coefficients reported) were used. (*N* = 63,652 for females, *N* = 53,387 for males). Abbreviations for Figs. 2–4 and 6 are given in 'Abbreviations'. Source data are provided as a Source Data file.

We used males as a negative control since men cannot experience the effects of being exposed to the stress test of pregnancy. Therefore, similar results between men and women would be indicative of bias, such as due to confounding by sociodemographic (e.g. education attainment) and biological (e.g. infertility) factors, rather than by an effect of repeated exposure to pregnancy. When using number of children in males as a negative control, we observed that associations for leucine, histidine, pyruvate, and ketone bodies were similar between men and women (i.e. directionally consistent, similar effect estimates and 95% confidence intervals overlapped between male and female estimates). On the other hand, association estimates for fatty acids, apolipoprotein A1, and lipoprotein-related traits were weaker or consistent with the null, and glycine was in opposite direction, in males compared to females (Fig. 4). For the MR analyses, we selected 32 SNPs

as instruments for parity, which explained 0.2% of its phenotypic variance with a corresponding mean F statistic of 31 (Supplementary data 8). It is unclear whether estimates from multivariable regression and MR analyses are consistent with each other due to the high level of uncertainty in the latter (Fig. 3 and Supplementary Fig. 7), which persisted even when using the larger sample of women with selected biomarkers assayed by clinical chemistry (Fig. 5).

### Age at natural menopause

In the main multivariable regression analyses (adjusting for age at baseline, education and body composition at age 10), older age at menopause was related to higher glycine, PUFA (e.g. DHA and LA), albumin, apolipoprotein B and sphingomyelins, but lower concentration of MUFA, pyruvate, acetoacetate, creatinine and GlycA

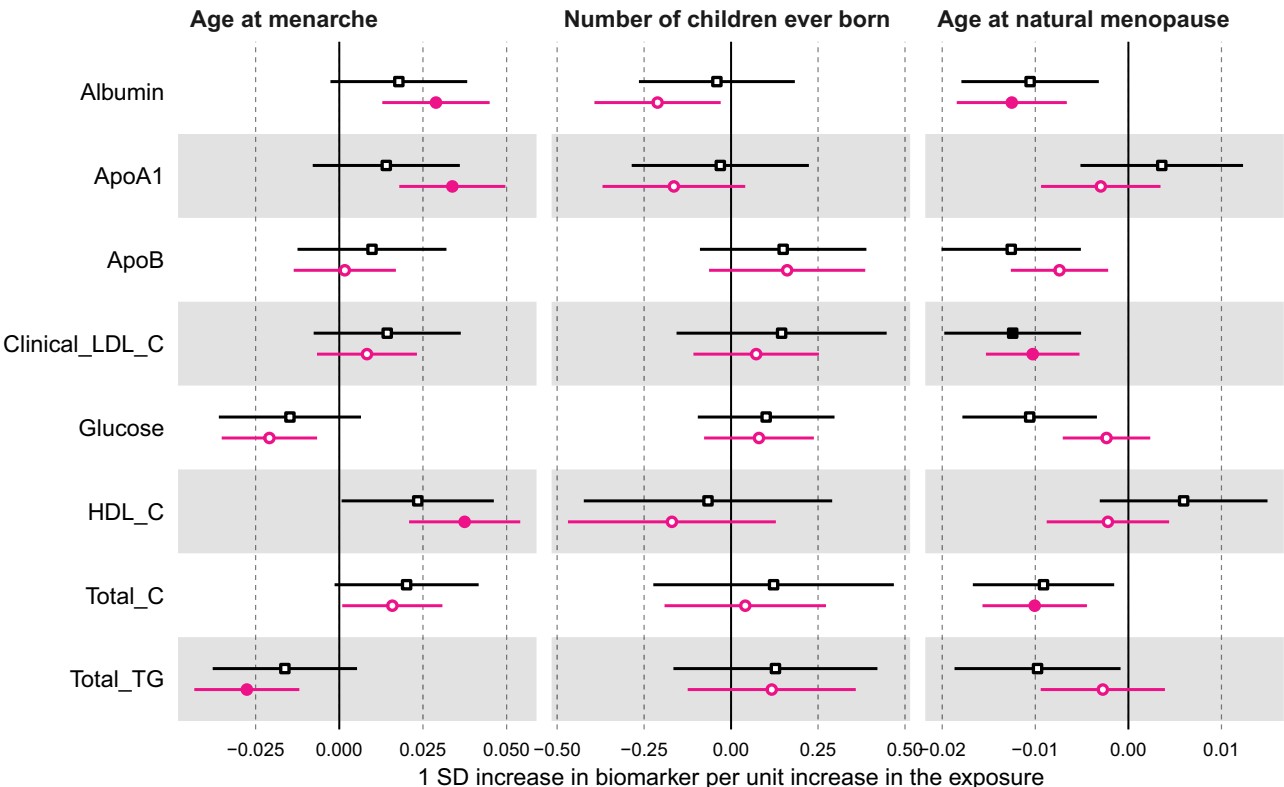

**Fig. 5 | Mendelian randomisation estimates for the relation between older age at menarche, number of children ever born, older age at menopause - and metabolic measures among females measured using NMR metabolomics (black, squares) or clinical chemistry methods (pink, circles).** Results are mean differences presented as standard unit changes in metabolic measure per 1 year increase in age at menarche, 1 additional birth, and 1 year increase in age at natural menopause, respectively. Circles/squares denote the mean differences and indicate *p*-value < 0.00093 (filled circles/squares) or ≥0.00093 (hollow circles/squares). Horizontal bars denote 95% confidence intervals. Mendelian randomisation models were estimated using the inverse variance weighted method for *N* = 62,209 for NMR metabolomics and *N* = 239,803 for clinical chemistry measures. Source data are provided as a Source Data file.

($P < 0.00093$) (Fig. 6 and Supplementary data 11). Older age at menopause was also associated with numerous lipoprotein-related traits at $P < 0.00093$, particularly with higher number of particles and lipid content in LDL, larger size of HDL particles, and lower size of VLDL particles (Fig. 6). The associations between age at menopause and LDL-related traits were observed across LDL subclasses (i.e. from small to large), whereas associations with HDL-related traits were mostly driven by larger HDL particles (Supplementary Fig. 11). In sensitivity analyses with further adjustments for BMI, smoking and alcohol status at baseline, associations between older age at natural menopause and metabolites remained similar, except for associations with HDL-related traits which were partly attenuated (Supplementary Fig. 12). There was evidence of non-linearity across 24 metabolites (Supplementary data 6 and Supplementary Fig. 13) in the multivariable regression when menopause was categorised (<49, 49–50, 51–53, >53 years). Restricted cubic spline models (with 4 knots) were generally consistent with the main analysis (assuming a linear association) until age at menopause -55 years when most metabolites decreased (Supplementary data 12 and Supplementary Fig. 14).

For the MR analyses, we selected 290 SNPs as instruments for age at natural menopause, which explained 8.2% of its phenotypic variance with a corresponding mean F statistic of 141 (Supplementary data 8). Estimates from multivariable regression and MR analyses were inconsistent in direction for many metabolic measures (Fig. 6). In particular, in contrast to results from multivariable regression, MR analyses indicated older age at menopause to be related to lower concentration of fatty acids (e.g. LA), albumin, apolipoprotein B, as well as lower

number of particles and lipid content in LDL across subtypes (from small to large) (Fig. 6 and Supplementary Fig. 11). For some metabolites, such as GlycA and HDL-related traits, results were consistent in direction between multivariable regression and MR. For alanine, glutamine and glucose, MR analysis suggested older age of menopause to be related to lower circulating metabolite levels, which had not been observed in multivariable regression analysis (Fig. 6 and Supplementary data 11). As expected, there was more uncertainty in MR estimates and only results for glutamine and some LDL- and VLDL-related measures passed the threshold for multiple test correction ($P < 0.00093$). Repeating the MR analyses in the larger sample of women ($N = 216,514–239,803$) with selected biomarkers assayed by clinical chemistry confirmed that older age at natural menopause was related to lower albumin, LDL-cholesterol, and total cholesterol at $P < 0.00093$ (Fig. 5).

We performed further analyses to investigate reasons underlying discrepant findings between multivariable regression and MR estimates for some metabolic measures (see Methods: Additional analyses for age at natural menopause: exploring the role of medication and chronological age). These analyses were restricted to the eight clinical chemistry biomarkers matching measures in the NMR platform to maximise statistical power since they have been measured in the full UK Biobank sample. First, we hypothesised that discrepant findings were related to differences in the sample used for multivariable regression, which excludes women with missing data on age at menopause because they had yet to go through it or had a surgical menopause (hereafter 'selected sample'), and two-sample MR, which

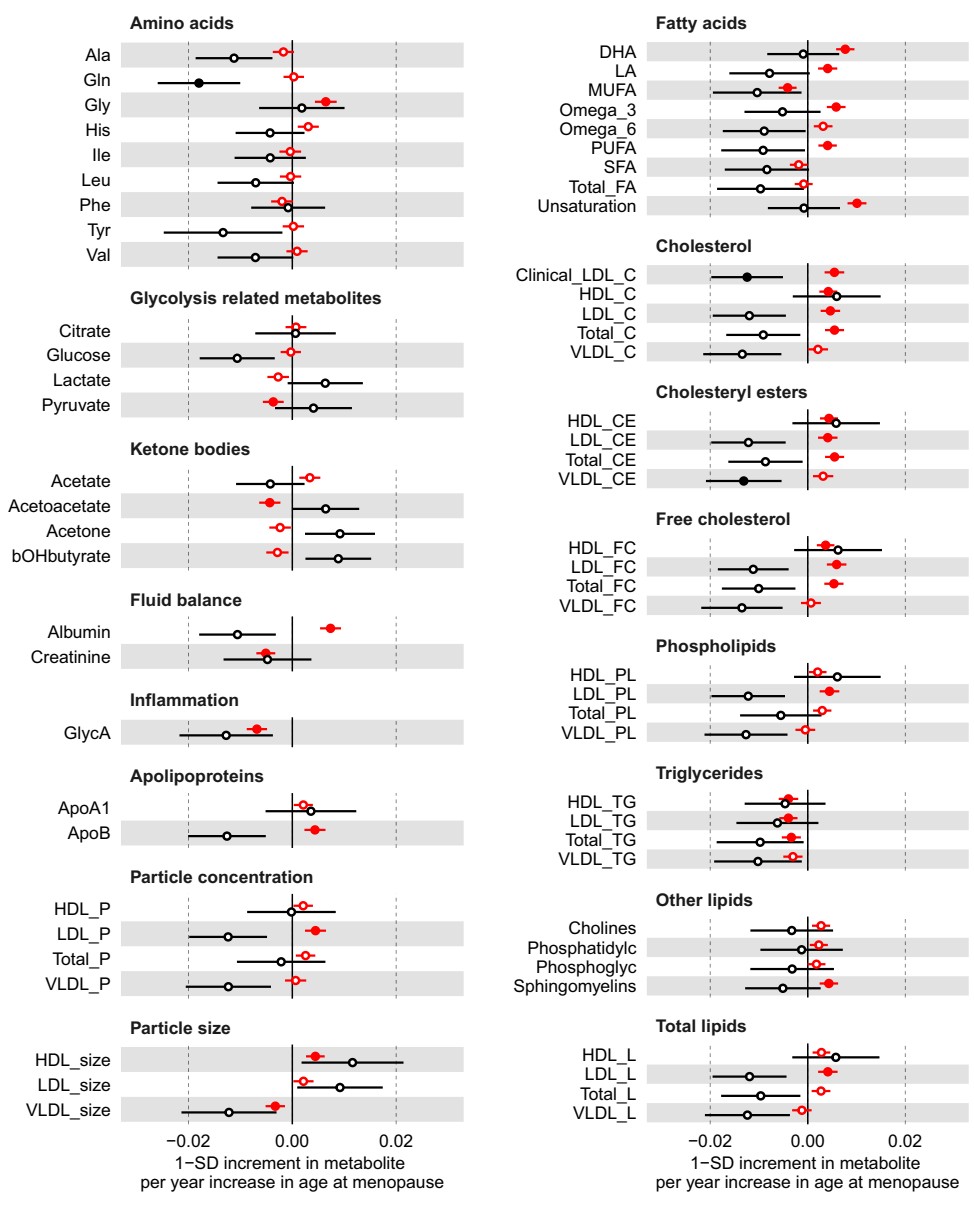

● MV    ● MR

**Fig. 6 | Multivariable regression (red) and Mendelian randomisation (black) estimates for the associations between older age at natural menopause and metabolic measures.** Results are mean differences presented as standard unit changes in metabolic measure per 1 year increase in age at menopause. Circles denote the mean differences and indicate p-value < 0.00093 (filled circles) or ≥0.00093 (hollow circles). Horizontal bars denote 95% confidence intervals.

Multivariable regression models (ordinary least squares, two-sided regression coefficients reported) were adjusted for age at recruitment, body size at age 10 and education (N = 36,253). Mendelian randomisation models were estimated using the inverse variance weighted method (N = 62,209). Abbreviations for Figs. 2–4 and 6 are given in 'Abbreviations' at the end of the manuscript. Source data are provided as a Source Data file.

includes women even if they are missing data on age at natural menopause (hereafter 'full sample'). To test that, we compared estimates from multivariable regression on the selected sample to MR on both the selected sample and full sample. In agreement with our hypothesis, multivariable regression and MR estimates for LDL-cholesterol and related traits are comparable when restricting to the selected sample. In contrast, for albumin, discrepant results were related to differences between multivariable regression and MR rather than between selected and full sample (Supplementary Fig. 15). Second, given women with missing data on age at menopause are typically pre-menopausal and younger, we explored age-stratified multivariable and MR estimates, which revealed a strong effect modification by chronological age on the association of age at menopause with LDL-

cholesterol and related traits – e.g. older age at menopause is related to substantially lower LDL-cholesterol in younger women (≤50 y) (e.g. multivariable regression (MV): −0.018 SD, 95% CI: −0.021, −0.015), but slightly higher LDL-cholesterol in older women (>63 y) (e.g. MV: 0.004 SD, 95% CI: 0.003, 0.006) (Supplementary Fig. 15). Differences related to chronological age at baseline were also observed for other biomarkers, such as albumin. When excluding women taking statins at baseline, we observed that the association between age at menarche and LDL-cholesterol estimated by multivariable regression was partly attenuated (−0.001 SD, 95% CI: −0.002, 0.000). However, excluding women using statins or hormone replacement therapy (HRT) at baseline did not substantially altered the chronological age patterned results (Supplementary Fig. 16).

## Exploring the plausibility of MR assumptions

We conducted a series of sensitivity analyses to explore the plausibility of key MR assumptions, required for the method to provide a valid test of the presence of a causal effect.

First, we tested whether MR findings are likely to be biased by population stratification, assortative mating and indirect genetic effects of parents using two approaches: (i) performing two-sample MR analyses using (sex-combined) data from a recent within-siblings GWAS, and (ii) conducting two-sample MR on negative control outcomes (i.e. skin colour and skin tanning ability). Two-sample MR estimates for the effect of genetic susceptibility for older age at menarche, parity, and age at natural menopause on five available biomarkers was broadly consistent when estimated among unrelated individuals or between siblings. Results for age at menarche were slightly overestimated for triglycerides and underestimated for glycated haemoglobin in unrelated individuals, while results for a positive relation between age at natural menopause and HDL-cholesterol was supported by analyses between siblings but not among unrelated individuals (Supplementary Fig. 17). We did not observe an association of genetically-predicted reproductive markers with skin colour or tanning (Supplementary data 13). Taken together, these sensitivity analyses indicate that our main MR estimates are unlikely to be substantially biased by population stratification, assortative mating, or indirect genetic effects of parents.

Second, we explored the presence of bias due to pleiotropic variants by using MR methods other than IVW: the weighted median estimator and MR-Egger. These methods can provide valid tests for the presence of a causal effect under different (and weaker) assumptions about the nature of the underlying horizontal pleiotropy compared to IVW. Estimates from IVW and weighted median were consistent in direction for most relationships between reproductive markers and metabolic measures. In most instances, estimates from MR-Egger method were uninformative given the high degree of uncertainty (Supplementary Figs. 18–20).

Third, we assessed potential bias due to sample overlap from including UK Biobank individuals in genetic association estimates for both exposures and outcomes. This was achieved by using data from previous GWAS that did not include UK Biobank, available for age at menarche and age at natural menopause, to select SNPs (and genetic associations estimates with exposures) for two-sample MR analyses (Supplementary data 14). When using SNPs selected from previous GWASes that did not include UK Biobank participants, results for of age at menarche and age at natural menopause were largely consistent, although less precise, compared to estimates from the main analyses using data with overlapping samples (Supplementary Figs. 21 and 22).

## Discussion

Our findings indicate that reproductive markers across women's lifespan are associated with distinct metabolic signatures in later life. Age at menarche, parity and age at natural menopause were related to numerous metabolic measures, representing multiple dimensions of metabolism, including amino acids, fatty acids, glucose, ketone bodies, and lipoprotein metabolism.

### Age at menarche

Age at menarche is frequently used as a proxy of puberty onset among females in epidemiological studies[2,28]. Our findings for the relation of age at menarche with metabolic measures were broadly concordant between multivariable regression and MR analyses, and were supportive of older puberty onset being related to a less atherogenic metabolic profile among adult women.

Both multivariable regression and MR estimates were markedly attenuated when accounting for adult BMI for most metabolic measures with few exceptions (e.g. glutamine and glycine), suggesting that the effect of age at menarche on adult metabolites are largely explained by adult BMI. There is evidence of a bidirectional relationship between puberty timing and adiposity, where pre-pubertal adiposity influences puberty timing, which in turn influences post-pubertal adiposity[13,28,29]. In addition, genetic variants influencing age at menarche are known to influence BMI before and after puberty[28,29].

The complex relationship between puberty timing and adiposity complicates inferences of the effect of age at menarche on the metabolic profile or disease risk in adulthood since the observed associations could reflect adult BMI mediating the effect of early age at menarche on metabolic measures or a confounding path from pre-puberty BMI. A previous one-sample MR study[28] investigating the effect of age at menarche on NMR metabolic measures reported that results were largely attenuated when accounting for BMI at 8 years old, which suggests that the estimated effect of age at menarche on the metabolic profile is largely confounded by pre-pubertal adiposity, though larger MR studies with repeat BMI and metabolic profiles before and after menarche are needed to rule out a potential causal mediated effect (i.e. older age at puberty resulting in higher BMI and, as a result, an atherogenic metabolic profile). In our study, accounting for self-reported adiposity in childhood in multivariable regression models did not substantially change effect estimates. This discrepancy might be related to residual confounding in our study (e.g. due to higher measurement error in our measure of childhood adiposity) or different age distributions between ours (mean = 55 years) and this previous study (mean = 18 years).

### Parity

Pregnant women undergo marked changes in physiology (e.g. lipid/glucose metabolism, adiposity, vascular function, hormone levels, and inflammatory response) and lifestyle (e.g. diet and physical activity[30]), most of which return to their pre-pregnancy state after delivery[20,30]. However, there are concerns that some of these changes might persist and accumulate over multiple pregnancies, impacting women's cardiovascular health in the future, or that pregnancy acts as a stress test, unmasking an underlying high risk for cardiovascular disease[30,31]. We used parity as a marker of being exposed to the physiological stress of multiple pregnancies.

In multivariable regression analyses, we found that higher parity, defined as number of children ever born, was associated with unfavourable changes in the metabolic profile (e.g. higher number of particles and lipid content in VLDL). Evidence from MR analyses is uncertain due to the high imprecision in effect estimates. Using males as a negative control, we showed that the associations between number of children ever born and metabolic measures among men were largely null for lipoprotein-related measures. This inconsistency between female and male findings reinforces that the metabolic signature associated with parity among females is likely to reflect a causal effect of parity on the metabolome rather than spurious results due to confounding or selection bias (assuming confounding structures and selection mechanisms are similar between men and women). A possible mechanism is that higher parity leads to greater insulin resistance in pregnant women and subsequently increases the production and secretion of hepatic triglycerides, which can lead to an increased lipid content in VLDL particles.

In line with our findings, other studies have reported that higher parity is related to higher cardiovascular disease risk in women[32]. Negative control analyses (comparing associations of number of children in women and men) have been conducted previously in two UK cohorts, with one suggesting that associations with lipids and body composition in women may be due to confounding (as associations are similar in women and men)[33,34] and the second, the largest of these studies to date, and the only one to look at disease end points, finding evidence of a stronger association for risk factors and coronary heart disease in women than men suggesting parity itself has some influence

on cardiovascular disease risk[5], Furthermore, studies in women only that are able to control for pre-pregnancy measures, suggest pregnancy and parity have a potentially lasting effect on adverse lipid profiles[35,36].

## Age at natural menopause

Observational studies suggest that menopause is associated with a worse cardiometabolic profile over and above chronological aging.[21,37–41] Previous cross-sectional[39] and longitudinal studies[21,40,42] indicate that the menopause transition is associated with a shift towards a more atherogenic lipoprotein profile, such as characterised by higher concentration of apolipoprotein B and LDL-cholesterol, and possibly higher circulating glucose[40] and inflammatory markers[21,39,40]. In addition, females experience a marked change in their metabolic profile at the age of late 40s and early 50s, which is not observed in males[39], providing further support for a role of menopause.

In our study, we focused on age at natural menopause as an indicator of reproductive aging. Findings from multivariable regression and MR were supportive of older age at menopause being related to lower systemic inflammation, as indicated by GlycA. On the other hand, MR indicated that older age at menopause is related to lower glucose and a less atherogenic lipoprotein profile (e.g. lower circulating apolipoprotein B and LDL-cholesterol) in line with previous studies, while multivariable regression did not support that. Multivariable regression results should be interpreted with caution as it was not possible to include ~40% of women, who did not have data on age at natural menopause, mostly due to being premenopausal (25%) or having a surgical menopause (12%). In addition, multivariable regression estimates were attenuated when excluding women reporting statins intake at baseline. In sensitivity analyses, we observed that multivariable regression and MR estimates are fairly consistent when stratified by chronological age, with older age at menopause being related to lower LDL-cholesterol in younger women (≤50 y) but slightly higher LDL-cholesterol in older women (>58 y).

Previous longitudinal studies indicated that LDL-cholesterol[40] and related traits increase sharply through the menopause transition and early postmenopausal years and then plateau with increasing postmenopausal years[43] This is in line with our cross-sectional analyses, in which we observed a non-linear pattern for several metabolites, such that mean metabolite levels increase linearly with age at menopause until 50–55 years old and then decline. Taken together, these findings might explain the pattern by chronological age in the association between timing of menopause and LDL-related traits. Menopause is a continuous dynamic process of progressive decline in ovarian function and circulating oestrogen levels. Therefore, we speculate that, among younger women (≤50 y) at baseline, those reporting a younger age at menopause are more likely to have fully experienced the menopause transition at the study baseline compared to those reporting an older age at menopause who may still be perimenopausal, which could explain the association between older age at menopause and lower LDL-related traits in this younger age group. On the other hand, among women older than 58 years at baseline, most will have experienced the full menopausal transition; in this age group, women reporting younger age at menopause will have been postmenopausal for many years, while women reporting older at menopause might still be in their early postmenopausal years, which might explain the association between older age at menopause and slightly higher LDL-related traits in this age group. In addition, we hypothesised these results could be related to higher intake of medications among older women, however, the chronological age-patterned results did not change substantially when excluding women reporting using statins or HRT at baseline. We note that such findings should be interpreted with caution given the lack of granularity in how we defined statins and HRT treatment and the potential for collider stratification bias[44]. Although these results are intriguing, larger longitudinal studies with longer follow-up will be needed to tease apart the complex nature, and possible time-varying effect of reproductive aging on metabolic profiles.

The largest two-sample MR analysis to date indicate that older age at menopause is related to lower risk of type 2 diabetes in females, but no difference in risk of cardiovascular disease or dyslipidemia (data combining males and females)[45]. This is in agreement with our MR analyses suggesting older age at natural menopause is related to lower glucose, and with evidence from randomised controlled trials of oestrogen therapy pointing to a protective effect on type 2 diabetes but no change in risk of cardiovascular diseases[46–48]. The mechanisms underlying the putative protective effect of older menopause on the risk of metabolic diseases in MR studies is unclear, but might reflect an effect of exposure to sex hormones or of slower cell aging, given genetic variants associated with age at natural menopause are highly enriched for genes in DNA damage response pathways[45,49]. The consistent results between MR of age at menopause and randomised controlled trials of oestrogen therapy for type 2 diabetes indicates that prolonged exposure to sex hormones is likely to be involved. Moreover, oestrogen regulates LDL particle receptor and clearance from the circulation[50,51], meaning that the sharp decrease in levels of oestrogen during menopause could plausibly explain some differences observed between menopause and lipoprotein traits[41].

## Strengths and limitations

To the best of our knowledge, this is the largest study to examine the long-term impact of key events in reproductive life on the multiple metabolic measures in women. The use of large-scale metabolomics data and the integration of multiple analytical approaches are key strengths of our study as these allowed us to strengthen the inference of the causal impact of these reproductive markers on the metabolic health of females.

It is important to note that the validity of our findings rely on the plausibility of the assumptions underlying each analytical approach. For multivariable regression, we cannot exclude the possibility of bias due to residual confounding, especially given we were unable to adjust for key confounders in multivariable regression as measures of these were not available at or before the exposure to reproductive factors. For the use of negative controls, we rely on the unverifiable assumption that residual confounding and selection bias are similar in females and males analyses. It is plausible that factors relating to metabolites, such as age, ethnicity, socioeconomic position, and BMI, relate similarly to number of children in females and males and hence that confounding structures are similar. For MR, we have conducted extensive sensitivity analyses supporting the validity of our results; however, we cannot rule out the possibility of bias due to violations of the core instrumental variable assumptions. In addition, MR analyses for parity were uninformative given the low proportion of phenotypic variance explained by the genetic instruments.

When assessing non-linearity, our multivariable regression results were generally consistent between the main analysis model (assuming a linear relationship) and categories for most metabolites. For metabolites that showed evidence of non-linearity, many seemed to plateau and decrease with older ages of menarche and menopause and similarly with higher parity (however, this was also where we had the least amount of data, which could be driving some of the non-linearity). We were unable to fit non-linear associations in an MR framework given this would require much larger sample sizes; future studies with larger sample sizes should be better powered to examine potential non-linear effects using MR and contrast those found in the multivariable regression.

Whilst some key sources of bias may remain in each method, a key strength of our study is exploring and focusing on results that are consistent across the different methods. As the sources of bias differ between the methods causal inference is strengthened where there is consistency, as we see, for example, in associations between

multivariable regression and MR for age at menarche, multivariable regression and negative control analyses for parity and multivariable regression and MR for age at natural menopause in relation to HDL-related measures and GlycA (but not LDL-related and other measures).

Across all analytical approaches, we cannot discard the presence of selection bias from using UK Biobank data given the low recruitment rate of the study (5%) and inclusion of healthier/wealthier individuals compared to the general UK population[52]. In addition, the metabolic traits measured by the NMR metabolomics platform cover a limited set of metabolic pathways[53], and, therefore, future studies including data from more sensitive metabolomics techniques, such as mass spectrometry, may improve coverage of the metabolome and provide insights into additional biological processes related to reproductive events. Triangulating results across different methods is useful for causal inference and where there are discrepant results it is important to explore these. We have found that the discrepant results between MR and multivariable regression for the association of age at menopause with some of the metabolites (notably LDL-cholesterol and related metabolites) are due to the exclusion of women with missing data on age at menopause in multivariable regression and a potential effect modification by chronological age in the association between age at menopause and some metabolic measures. However, we acknowledge we lack power to fully explore the mechanisms for this given the current number of UK Biobank participants with NMR data.

Overall, we found supportive evidence that reproductive factors may affect females' metabolic profile later in life. Evidence supporting a relation between later pubertal timing and a less atherogenic metabolic profile was largely explained by adult BMI, but studies with repeat assessment of metabolites and BMI are necessary to determine whether this reflects confounding by BMI or a causal effect of age at puberty that is mediated by BMI. Findings linking higher parity to a more atherogenic profile were supported by the negative control analyses but imprecisely estimated in Mendelian randomisation. Evidence supporting a relation between slower reproductive aging and a less atherogenic metabolic profile was mostly observed among younger women. These results could contribute to identifying novel markers for the prevention of adverse cardiometabolic outcomes in women and/or methods for accurate risk prediction.

## Methods

This study was approved under UK Biobank Project 30418 and 81499. UK Biobank received ethical approval from the Research Ethics Committee (REC reference for UK Biobank is 11/NW/0382).

### Study participants
UK Biobank is a population-based cohort consisting of approximately 500,000 men and women recruited between 2006 and 2010 from across the UK (age range at recruitment: 38 years to 73 years old)[54]. UK Biobank participants have provided a range of information via questionnaires and interviews, including on sociodemographic, lifestyle, health, and reproductive factors; as well as biological samples and physical measures (data available at www.ukbiobank.ac.uk). A subset of approximately 20,000 were selected for repeat assessment between 2012 and 2013. A full description of the study design, participants and quality control (QC) methods have been described in detail previously[55].

### Reproductive traits
Women were asked a detailed set of questions about their reproductive health via a self-reported questionnaire. Parity was based on the number of live births reported whilst in men number of children were reported. Age at menarche and age at natural menopause were reported in years. Age at natural menopause therefore excluded women who had not yet gone through the menopause ($N = 25,740$) because they were premenopausal ($N = 15,418$) or who had a surgical

menopause or other reason ($N = 10,322$). (Table 1, Supplementary data 3).

### NMR metabolic measures
Metabolic traits were measured using a targeted high-throughput NMR metabolomics (Nightingale Health Ltd; biomarker quantification version 2020)[56]. This platform provides simultaneous quantification of 249 metabolic measures, consisting of concentrations of 165 metabolic measures and 84 derived ratios, encompassing routine lipids, lipoprotein subclass profiling (including lipid composition within 14 subclasses), fatty acid composition, and various low-molecular weight metabolites such as amino acids, ketone bodies and glycolysis metabolites. Technical details and epidemiological applications have been previously reviewed[18,53]. Pre-release data from a random subset of 126,846 non-fasting plasma samples collected at baseline or first repeat assessment were made available to early access analysts. 121,577 samples were retained for analyses after removing duplicates and observations not passing quality control (QC) (i.e. sample QC flag Low protein, biomarker QC flag Technical error, or samples with insufficient material). All metabolic measures were standardised and normalised prior to analyses using rank-based inverse normal transformation.

### Clinical chemistry measures
We used data on the eight biomarkers assayed using clinical chemistry techniques, as previously described[57], that matched measures in the NMR metabolomics platform − i.e. albumin, apolipoprotein A1, apolipoprotein B, glucose, HDL-cholesterol, LDL-cholesterol, total cholesterol, and triglycerides. These measures are available in most UK Biobank participants and were used in Mendelian randomisation analyses, as described under 'Statistical analyses', to increase statistical power and check agreement with results from NMR metabolic measures. All biomarkers were standardised and normalised prior to analyses using a rank-based inverse normal transformation.

### Summary data on genetic associations with metabolic measures
Genotype data was available for 488,377 UK Biobank participants, of which 49,979 were genotyped using the UK BiLEVE array and 438,398 using the UK Biobank axiom array. Pre-imputation QC, phasing and imputation are described elsewhere[58]. Genotype imputation was performed using IMPUTE2 algorithms[59] to a reference set combining the UK10K haplotype and HRC reference panels[60]. Post-imputation QC was performed as described in the "UK Biobank Genetic Data: MRC-IEU Quality Control" documentation[61]. Genetic association data for metabolic measures was generated using the MRC IEU UK Biobank GWAS pipeline[62]. Briefly, we restricted the sample to individuals of 'European' ancestry as defined by the largest cluster in an in-house k-means cluster analysis performed using the first 4 principal components provided by UK Biobank in the statistical software environment R ($n = 464,708$). Genome-wide association analysis (GWAS) was conducted using linear mixed model (LMM) association method as implemented in BOLT-LMM (v2.3)[63]. Population structure was modelled using 143,006 directly genotyped SNPs (MAF > 0.01; genotyping rate > 0.015; Hardy−Weinberg equilibrium $p$-value < 0.0001 and LD pruning to an r2 threshold of 0.1 using PLINKv2.00). Models were adjusted for genotyping array and fasting time and were restricted to the subsample of women.

### Covariables
For multivariable regression analyses, confounders were defined a priori based on them being known or plausible causal factors for reproductive traits and cardiovascular risk via higher/lower metabolites. A minimal set of adjustments were made in the main multivariable regression analyses as most confounders were not assessed prior to or around when the reproductive traits occurred. Specifically,

we adjusted for education as a categorical variable (University, A-levels, O levels (or equivalent) or other), age at baseline and retrospectively reported body size at age 10 (average, thinner, plumper) in all regression analyses. In additional analyses we also partially adjusted for the full set of defined confounders using baseline measurements (mostly after the occurrence of exposures) as correlates of the before exposure measures (see below in statistical analyses).

## Statistical analyses

We used multiple approaches (i.e. multivariable regression, negative control and MR) relying on different assumptions to explore the causal role of reproductive traits on later life metabolic profile. Information on 'Statistics & Reproducibility' is provided at the end of this section. All analyses were conducted using Stata16 (StataCorp, College Station, TX) or R 4.1.1 (R Foundation for Statistical Computing, Vienna, Austria) and results presented as differences in means for each metabolic trait in standard deviation (SD) units per 1 child difference for number of children and per 1 year difference for age at menarche and age at menopause, facilitating the comparison of results from different methods.

For both multivariable and MR analyses, we corrected for multiple testing using the Bonferroni method considering 3*18 = 54 independent tests ($P = 0.05/54 \approx 0.00093$). This was based on the three exposures included in our analyses (i.e. age at menarche, parity, and age at natural menopause) and the 18 independent features explaining over 95% of variance in the highly correlated NMR metabolic measures in our dataset as estimated by principal component analysis[64].

**Multivariable regression.** In the main analyses we used linear regression, with three sets of models: (1) no adjustments, (2) adjusted for education, age at baseline and body composition at age 10 and (3) model (2) additionally adjusted for baseline variables collected at the first assessment at (mean) age 56 years (SD = 8) including BMI, smoking and alcohol status. By adjusting for the baseline variables at the first assessment we can either block the confounding path or create bias if these variables are mediators. If the results change between model (2) and (3) it is hard to distinguish whether its the correct adjustment for confounding or whether it is a mediated path. Because of this we considered model (2) to be the best causal estimate and present models (1) and (3) in supplementary material. For age at menarche, education will have been measured after the exposure. However, as it is influenced by parental education, income and occupation (occurring before menarche), and unlikely to be determined by age at menarche, we a priori considered it as a proxy of early life[65]. In sensitivity analyses we assessed whether there was a non-linear relationship between each reproductive trait and non-derived metabolites. For ease of presentation, we excluded measures that were derived (eg ratios) or related to lipoprotein subfractions as these are highly correlated with one or more of the 55 non-derived metabolites. We compared the categorised reproductive trait entered into the model as a categorical variable and as a continuous variable using a likelihood ratio test. Age at menarche and age at menopause were categorised into tertiles (<13, 13–14, >14 years) and quartiles (<49, 49–50, 51–53, >53 years), respectively. Parity was categorised as 0, 1, 2, and 3+. Results were plotted against the first reference category and the p-value for linear trend reported. For any metabolites that showed evidence of non-linearity, restricted cubic splines (with either 3, 4, or 5 knots placed at percentiles as suggested by Harrell[66] for each reproductive trait) were fit and compared to the main analysis model (assuming a linear association) using AIC (BIC and root mean square error also shown).

**Negative control analyses.** Negative control analyses aim to emulate a condition that cannot involve the hypothesised causal mechanism but is likely to have similar sources of bias that may have been present in the association of interest[5,27]. We used males as negative controls to assess potential biases in the association between parity (proxied by number of live births) and metabolic measures in women. If associations between number of live births and metabolic measures in women reflect a causal effect of parity on women's metabolic health, one would expect number of live births to be associated with metabolic measures in women but not in men, given men do not experience pregnancy. Similar to the multivariable regression analyses, we test the association between number of children (men) and their measured metabolites and present three sets of models: (1) with no adjustments, (2) adjusted for education, age at baseline and retrospectively self-reported body composition at age 10 and (3) model (2) additionally adjusted for baseline variables collected at the first assessment at (mean) age 56 years (SD = 8) including BMI, smoking and alcohol status.

**Mendelian randomisation.** We used two-sample MR to explore the effect of older age at menarche, higher parity, and older age at natural menopause on women's metabolic profile. Publicly available GWAS summary data were used for SNP-reproductive traits associations (sample 1) and UK Biobank summary GWAS data for SNP-metabolite associations (sample 2). This approach does not require all participants to have data on both exposure and outcome, and, therefore, allows us to retain the largest possible sample sizes, meaning that power to detect a causal effect is increased[67].

## Selection of genetic instruments

**Age at menarche.** Genetic instruments were selected from a GWAS of age at menarche, which included 329,345 women of European ancestry (Supplementary data 14)[68]. Linear regression models were used to estimate the association between genetic variants and age at menarche (in years) adjusting for age at study visit and study-specific covariables. For our analyses, we selected the 389 independent SNPs reported by the GWAS to be strongly associated with age at menarche ($P$-value < $5*10^{-8}$) in the discovery metanalyses. Given the age at menarche GWAS included UK Biobank participants (maximum estimated sample overlap: ~20%), we have also selected an additional set of age at menarche-associated genetic variants ($N = 68$ SNPs) using data from a previous GWAS that did not include UK Biobank (details in 'Sensitivity analyses' below and Supplementary data 14)[69].

**Parity.** Genetic instruments were selected from a GWAS of number of children ever born, as a proxy of parity, which included 785,604 men and women of European ancestry from 45 studies (Supplementary data 14)[70]. Number of children ever born was treated as a continuous measure and included both parous and nulliparous women. Linear regression models were used to estimate the association between genetic variants and number of children ever born adjusting for principal components of ancestry, birth year, its square and cubic, to control for non-linear birth cohort effects. Family-based studies controlled for family structure or excluded relatives. The sex-combined metanalysis also included interactions of birth year and its polynomials with sex. For our analyses, we used the 32 independent SNPs reported by the GWAS to be strongly associated with number of children ever born ($P$-value < $5*10^{-8}$) in either the sex-combined (28 SNPs) or female-specific (4 SNPs) metanalyses and summary association data from the female-specific metanalyses. The GWAS included UK Biobank (maximum estimated sample overlap: 14%).

**Age at natural menopause.** Genetic instruments were selected from a GWAS of age at natural menopause conducted in 201,323 women of European ancestry (Supplementary data 14)[17]. Linear regression models were used to estimate the association between genetic variants and age at natural menopause (in years). For our analyses, we selected 290 SNPs reported by the GWAS to be strongly associated with age at natural menopause ($P$-value < $5*10^{-8}$). Where available, we used

association data from the sample combining discovery and replication stages (*N* = 496,151). Given the age at menarche GWAS included UK Biobank participants (maximum estimated sample overlap: 13% considering the GWAS combined discovery and replication samples), we have also selected an additional set of age at natural menopause-associated genetic variants (*N* = 42 SNPs) using data from a previous GWAS that did not include UK Biobank (details in 'Sensitivity analyses' below and Supplementary data 14)[49].

## Main analyses

We used a standard two-sample MR method, the inverse variance weighted (IVW) estimator, to explore the effect of age at menarche, parity and age at natural menopause on women's metabolic profile by combining genetic association estimates for reproductive traits (extracted from published GWASes data) with genetic association estimates for the metabolic measures (generated from UK Biobank data). Given a priori evidence of a potential bidirectional relationship between age at menarche and BMI, we also used multivariable IVW to test the effect of age at menarche on metabolic measures accounting for adult BMI. For multivariable IVW analysis, apart from the data previously described, we used summary genetic association data for BMI extracted from the 2015 metanalysis by the GIANT consortium (*N* = 339,224 individuals not including UK Biobank participants)[71].

## Sensitivity analyses

Several sensitivity analyses were conducted to explore the plausibility of the three core MR assumptions, which are required for the method to provide a valid test of the presence of a causal effect.

Assumption 1: the genetic instrument must be associated with the reproductive trait. We selected genetic variants reported to be strongly associated with reproductive in the largest available GWAS and estimated the proportion of phenotypic variance explained ($R^2$) and F-statistics for the association of SNPs with reproductive traits among females as an indicator of instrument strength.

Assumption 2: the association between genetic instrument and outcome is unconfounded. One of the main motivations for using MR is to avoid unmeasured confounding. However, there is growing evidence that, in some instances, MR studies can be confounded when using data from unrelated individuals due to population stratification, assortative mating and indirect genetic effects of parents[72,73]. We used two approaches to explore whether these were likely to bias our main results. First, we used sex-combined data from a recent within-sibship GWAS, including up to 159,701 siblings from 17 cohorts, to test the effect of genetic susceptibility to higher age at menarche, parity and age at menopause on metabolic markers (i.e. LDL-cholesterol, triglycerides, HDL-cholesterol, C-reactive protein, and glycated haemoglobin)[72]. C-reactive protein and glycated haemoglobin were used as proxies for inflammation and hyperglycaemia, respectively, given GlycA and glucose were not available. Within-sibling MR designs control for variation in parental genotypes, and so should not be affected by population stratification, assortative mating and indirect genetic effects of parents[72–74]. Second, we performed IVW on negative control outcomes (i.e. skin colour and skin tanning ability) since these could not conceivably be affected by the exposures and any evidence for an association between reproductive traits and, these negative control outcomes would be indicative of residual population stratification in the exposure GWAS[75].

Assumption 3: the genetic instrument does not affect the outcome except through its possible effect on the exposure. A key violation of this assumption is known as horizontal pleiotropy, where genetic variants influence the outcome through pathways that are not mediated by the exposure[76]. We explored the presence of bias due to horizontal pleiotropy by using other MR methods: the weighted median estimator and MR-Egger. These methods can provide valid tests of a causal effect under different (and weaker) assumptions about the nature of the underlying horizontal pleiotropy. The weighted median estimator requires that at least 50% of the weight in the analysis stems from valid instruments. The MR-Egger estimator assumes that the instrument strength is independent of its the direct effects on the outcome (i.e. INSIDE assumption).

In addition to the core assumptions, the two-sample MR approach assumes that genetic associations with exposure and outcome were estimated from two comparable but non-overlapping samples. We restricted our analyses to European adult individuals to ensure that samples were comparable. We assessed potential bias due to sample overlap by conducting MR using SNPs selected from previous GWAS of age at menarche and age at natural menopause that did not include UK Biobank (Supplementary data 14).

## Statistics & reproducibility

No sample size calculation was conducted for this analysis, as it was based on secondary data from UK Biobank. We used phase 1 data on metabolic traits assessed using a targeted high-throughput NMR metabolomics platform (Nightingale Health Ltd; biomarker quantification version 2020). Pre-release data from a random subset of 126,846 non-fasting plasma samples collected at baseline or first repeat assessment were provided, and after removing duplicates and observations not passing quality control (QC), 121,577 samples were retained for analyses. Duplicates (of those with repeated metabolite measures) were removed from our final dataset and therefore all analyses are based on independent observations. By definition we excluded women who had not yet gone through the menopause or who had a surgical menopause in the multivariable regression analyses for age at menopause. Whilst for the two sample mendelian randomisation this includes all women's genetically predicted age at menopause. As this is a cohort study exploring reproductive traits it is not possible randomise our exposures: age at menarche, parity, and age at menopause. However, in the regression analyses, based on prior knowledge and the data, we controlled for as many confounders as possible using multivariable regression. In the main analysis we adjusted for the following confounders: adjusted for education, age at baseline and body composition at age 10. By doing so, we aimed to ensure that our estimates of interest were influenced mainly by the effects of the reproductive traits we were studying, minimising potential bias from other factors. For the mendelian randomisation analyses, conceptually this can be thought of as a natural experiment where genetic variants instrumenting for exposures are randomly allocated at conception. Sensitivity analyses testing these assumptions were carried out. For example, in the multivariable regression we present and compare three sets of models each adjusting for different covariates. Blinding was not applicable in this cohort study as the exposure information (age at menarche, parity, and age at menopause) is inherent to the participants and is not influenced or altered by the study design (other than through selection as described).

## Additional analyses for age at natural menopause: exploring the role of medication and chronological age

We performed further analyses to investigate reasons underlying discrepant findings between multivariable regression and MR estimates for some metabolic measures. These analyses were restricted to the eight clinical chemistry biomarkers matching measures in the NMR platform to maximise statistical power since they have been measured in the full UK Biobank sample.

First, we hypothesised that discrepant findings were related to differences in the sample used for multivariable regression, which excludes women with missing data on age at menopause (hereafter 'selected sample'), and two-sample MR, which includes women even if they are missing data on age at natural menopause (hereafter 'full sample'). To test that, we compared estimates from multivariable

regression on the selected sample to MR on both the selected sample (one-sample MR) and full sample (two-sample MR).

Across analyses, we excluded women who were related, of non-European ancestry, or did not pass the quality control for the genetic data, as described in the MRC IEU QC pipeline - version 2 (https://doi.org/10.5523/bris.1ovaau5sxunp2cv8rcy88688v) ('full sample' = 208,062). For multivariable regression and one-sample MR, the sample was further restricted to the 'selected sample', consisting of women with complete data on age at menopause, education, age at baseline and body composition at age 10 (N = 123,278). For two-sample MR, all women, regardless of missing data on age at natural menopause, were included when estimating the instrument-outcome association, although women with missing data on age at natural menopause were inherently excluded when estimating the instrument-exposure association.

Multivariable regression models were adjusted for education, age at baseline and body composition at age 10. For the MR analyses, we derived a weighted polygenic score (PGS) for age at menopause including the SNPs and weights from the same GWAS used for the main analyses (Supplementary data 14). One-sample MR was performed using two-stage least square regression, accounting for the genotyping array and the top 40 principal components of ancestry. Two sample MR was performed using the Wald ratio estimator of the PGS-outcome association by the PGS-exposure association estimates adjusted by genotyping array and the top 40 principal components of ancestry.

These multivariable regression, one-sample and two-sample MR analyses were performed on the combined sample and on four sub-groups defined by chronological age (≤50 y, >50 y & ≤58 y, >58 y & ≤63 y, and >63y).

In addition, we repeated these analyses after excluding women reporting at baseline use of statins, derived from data field 20003 (codes: 1140861958, 1140888594, 1140888648, 1140910632, 1140910654, 1141146234, 1141192410) or hormone replacement therapy (HRT), derived from data field: 3546.

### Reporting summary
Further information on research design is available in the Nature Portfolio Reporting Summary linked to this article.

## Data availability
UK Biobank received ethical approval from the Research Ethics Committee (REC reference 582 for UK Biobank is 11/NW/0382). The NMR metabolomic data is available within UK Biobank The current analysis was approved under UK Biobank Project 30418 and 81499. Bonafide researchers can request access to UK Biobank data via the Access Management System (AMS). The age at menarche, age at natural menopause and number of children were obtained from https://www.nature.com/articles/ng.3841[68], https://www.nature.com/articles/s41586-021-03779-7[45], and https://www.nature.com/articles/s41562-023-01528-6[70], respectively. Please see Supplementary data 14 for more information, respectively. Source data are provided with this paper.

## Code availability
Analysis scripts and the analysis plan can be found on the following GitHub page: https://github.com/gc13313/nmr_repro. https://doi.org/10.5281/zenodo.10371915.

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

## Acknowledgements

We are extremely grateful to Nightingale Health Ltd for the use of their data and for their helpful discussions throughout. We want to acknowledge participants and investigators from UK Biobank and the multiple large-scale GWAS consortia which made summary data available. This work used the computational facilities of the Advanced Computing Research Centre, University of Bristol -http://www.bristol.ac.uk/acrc/. The current analysis was approved under UK Biobank Project 30418 and 81499. We are also grateful to Professor Kate Tilling (University of Bristol), Prof Zoltan Kutalik, and Leona Knusel (University of Lausanne) who helped us with additional analyses undertaken to explore discrepant results between multivariable regression and two sample MR for the association of age at natural menopause with biomarkers.

**Funding.** This research is supported by the University of Bristol and UK Medical Research Council (MRC) (MC_UU_00032/05, all authors), the European Union's Horizon 2020 research and innovation programme under grant agreement No 733206 LifeCycle (G.L.C. and D.A.L.), a University of Bristol Vice-Chancellor's Fellowship (M.C.B.), the British Heart Foundation (AA/18/7/34219, M.C.B. and D.A.L. and CH/F/20/90003, D.A.L.) and the UK National Institute of Health Research (NF-0616-10102, D.A.L.). The funders had no role in study design, data collection and analysis, decision to publish, or preparation of the manuscript. This publication is the work of the authors and all authors will serve as guarantors for the contents of this paper.

## Author contributions

D.A.L. conceived the study. D.A.L., M.C.B. and G.L.C. designed the study. M.C.B. and G.L.C. performed the analyses. M.C.B. and G.L.C. wrote the original draft of the manuscript with input from D.A.L. All authors were involved in the interpretation of results, helped refine the manuscript, and approved its final version.

## Competing interests

D.A.L. reports receiving support from several national and international government and charity research funders, and grants from Roche Diagnostics and Medtronic Ltd for work unrelated to that presented here. G.L.C. and M.C.B. declare that they have no competing interests.
