## [Peer Review File · Nature Communications]

REVIEWER COMMENTS

Reviewer #1 (Remarks to the Author):

What are the noteworthy results?

The authors use complementary multiple variable regression (MV) and genetic mendelian randomisation (MR) approaches to test the associations of 3 reproductive exposures, Menarche, Parity and Menopause, on NMR metabolomic outcomes in a large sample of women, mean age 56 years.

Does the work support the conclusions and claims, or is additional evidence needed?

MV and MR are indeed complementary approaches to support causal understanding. However, as expected "there was a higher degree of uncertainty for IVW estimates" and none of the MR results for Menarche or Parity reached significance (as shown in Figure 2B). So it is unclear how can the authors justify their summary, Abstract "Older age of menarche was related to a more atherogenic metabolic profile in (MV and) MR". Albeit, the resolution of Figure 2B is too poor to see the p-value threshold used for MR but it seems different to that for MV - please clarify and justify this threshold and consider the overall conclusion.

Turning to Menopause, here they find several opposing effects for MV and MR on Cholesterol parameters, IDL, large medium and small LDL, and very small VLDL - these are illustrated together in Suppl Fig 6. Despite a number of sensitivity tests, the reason for this discrepancy is not identified. Opposite MR results are still found in within-siblings, Weighted median and MR Egger, and using main set SNPs. So MR and MV then are not so complementary approaches. Without knowing the reason for the difference it is unclear which direction is the correct one.

One possible other difference between MV and MR is the selected population. MV is restricted to the 61% women who report reaching natural menopause. MR is in the all women. It is possible that the effects of menopause age differ somehow before and after menopause. This could be checked.

Other comments

P-value < 0.00076. This correction for 3 x 22 tests is based on a calculation of NMR metabolite data in a study of 14 cohorts in Ref 15. This threshold could easily be confirmed and calibrated to the current data.

Suppl Fig 4, results for parity, the very wide X-axis scale means it is impossible to tell if MV results are significant. I suggest you use a tighter scale and truncate the longer confidence intervals for MR which are almost all non-significant.

Glycated Haemoglobin is mentioned in the results text but I can't find it in the Figures.

Figure 1. What is meant by "(no residual confounding)"? Surely this condition cannot be met.

Reviewer #2 (Remarks to the Author):

In this paper, using a triangulation approach, evidence is sought for the mechanisms relating female reproductive risk factors to cardiometabolic traits. This is a very large and very well conducted study finding that reproductive risk factors are associated with metabolic changes later in life. However, I think the conclusion may be a bit optimistic, given the fact that so very few associations seem to match with multivariable regression and MR.

This paper clearly contributes to the existing literature, even though most of the findings are not new, this is the first paper to take such an extensive approach. For all analyses, potential biases are taken clearly into account, and all MR assumptions are checked.

I have a few comments on this paper:

- My most important comment would be on the fact that this paper includes so many results, and it is often very hard to get a clear understanding from the tables and figures. So, it would be helpful to include some tables, that summarize the most important findings a bit more. For instance, a table with an overview for each determinant what traits were overlapping with the two (or three) methods with the estimates in there, so that I can quickly compare the results for those traits that in the end are considered to be causally related to the exposures.

- The authors state that for the negative male analyses similar results in men and women would suggest that parity is not causally associated. I tend to disagree. It just means that it is not the pregnancy, but it could be due to just having kids. It is not being said that the SNPs identified for parity reflect pregnancy. In fact, one could question what kind of SNPs would come up with the number of children as an outcome, given that some of it is choice (many people choose to only have two children for instance),

some of it is infertility of either the woman or the men involved, etc. which may be similar in men and women. So what are we in fact measuring? Results show some differences between men and women, suggesting that there is some biological effect of the actual pregnancy. However, what keeps bothering me here is the additional effect of breast feeding, which is also associated with number of children, and the effect of that on BMI later in life. This in combination with the low power of this MR analysis, makes me wonder whether the authors might be a bit overoptimistic in their conclusions. Some of this needs to be incorporated in the discussion.

- Age at menarche: My main concern is that only linear associations were tested, while observational evidence suggests a non-linear association with cardiovascular disease. Might a non-linear analysis change the results?

- Age at menopause: Results are very much in line with literature. I would suggest to include the paper of de Kater et al, as in that study we were really able to take age out of the association and found an effect of menopause on lipid levels, independent of age.

Reviewer #3 (Remarks to the Author):

The authors analyzed the associations of age at menarche, menopause, and parity with 249 metabolic traits measured with nuclear magnetic resonance (NMR), utilizing the data from 65 487 UK Biobank women. The major strength of the paper, in addition to the large sample size, is leveraging evidence across 3 methods: multivariable regression analysis (MV), Mendelian Randomization (MR) and use of negative control (men) to reduce the confounding effects inherent in observational studies. Moreover, there's a paucity of well-conducted metabolomic studies in the reproductive field and the present study contributes important new evidence in this regard. The authors found that older age at menarche and menopause was associated with a more favorable metabolic profile in women and mixed results were obtained for parity. While the manuscript is overall very well written and the data and methodology are presented in a detailed way, I feel that some revisions are necessary to further strengthen the paper.

Abstract

1) The Abstract (p. 2) and Conclusions (p. 11) do not completely agree. While in the Abstract the authors claim that older age at menarche is related to a more proatherogenic profile (lines 28-29), in the Conclusions it is stated that “Overall, older age at menarche/menopause were [sic!] related to a more favorable metabolic profile” (lines 347-348). The authors should rephrase both parts and align the conclusions.

2) Overall, I recommend focusing on the results which were consistent/robust across MR and MV when drawing the conclusions.

Methods

3) Why weren't women who have never been pregnant used as a negative control? Since there is a plethora of sex-and gender-specific differences between women and men which might affect the metabolites and the associations with parity, including behavioral and biologic factors, involvement of inherent physiological differences, like the impact of the sex hormones, differences in cardiovascular risk factors, gender disparities from cultural and environmental factors, and differences between the sexes in the diagnosis and treatment of diseases, the authors might actually introduce new spurious confounding instead of controlling for it. The analyses with metabolites are cross-sectional, so the argument that some never-pregnant women might become pregnant in the future is less relevant.

4) Reported body size at age 10 (average, thinner and plumper) might not sufficiently capture the confounding nature of pre-exposure BMI, as these categories are imprecise, moreover, prone to recall bias. In fact, after considering the attained BMI at the time of metabolites measurement (post-exposure), most of the associations were attenuated (supplemental Figure 2) in MV and none survived the Bonferroni threshold in MR (Figure 2B) relating to parity or age at menarche.

5) Bonferroni correction: Could the authors please explain the rationale behind “This was based on the three exposures included in our analyses (i.e. age at natural menarche, parity, and age at natural menopause) and the 22 independent features estimated among the highly correlated NMR metabolic measures that explained over 95% of variance in these measures in an independent dataset using principal component analysis”, p. 14 lines 430-434. Since the authors chose an exploratory approach when examining the associations between metabolites and reproductive risk factors, I believe they should apply a more conservative method like correcting for all independent tests (all detected metabolites): $0.05/249 = 2.008 \times 10^{-4}$ or using an FDR correction based on all 249 tests.

6) Did the authors inspect the potential non-linearity between age at menarche/menopause and metabolites levels? This could be achieved with restricted cubic splines or fractional polynomials, for example. As a sensitivity analysis, I'd suggest a categorization of age at menarche and menopause.

Results

7) Table 1: why did the authors decide to present the characteristics stratified by sex, if the focus of the paper was female reproductive factors? Men are investigated only as negative controls for parity and are not the primary focus of the paper. It would be more sensible to present the data stratified by categories of (older) age at menarche, menopause, and parity (see also my comment regarding sensitivity analyses above).

8) According to figure 2B (MR analyses), no metabolic measure was significantly associated with age at menarche or parity, and as seen from Figure 3, the MV and MR analyses do not agree with each other for the associations with parity and age at menopause. Moreover, the % of variability in reproductive factors explained by the selected SNPs was rather low (ranging from 0.2 to 8%, supplemental Table 2), and taking together, these data suggest that the results should be interpreted with caution.

Discussion

9) Regarding the associations of specific metabolites with parity (p. 9), a more cautious interpretation of potential causality is warranted, since a) the results between MV and MR are not in agreement, b) by using males as a negative control group the authors might introduce new confounding through the above-discussed mechanisms (see my comment #3)

10) NMR has a relatively low sensitivity and, moreover, can detect lower number of metabolites as compared to LC-MS. The authors should address this limitation.

11) For the metabolites, robustly associated with reproductive risk factors across both the MV and MR analyses, can the authors maybe speculate about possible mechanistic pathways involved in those associations?

REVIEWER COMMENTS

We thank the editor and reviewers for their helpful responses. We have responded below, with reviewers' comments in blue and our responses in black. Where we have added new text, we have this in italics and then in the main text document tracked these changes.

Reviewer #1 (Remarks to the Author):

Reviewer 1 (point 1):

What are the noteworthy results? The authors use complementary multiple variable regression (MV) and genetic mendelian randomisation (MR) approaches to test the associations of 3 reproductive exposures, Menarche, Parity and Menopause, on NMR metabolomic outcomes in a large sample of women, mean age 56 years.

Does the work support the conclusions and claims, or is additional evidence needed? MV and MR are indeed complementary approaches to support causal understanding. However, as expected "there was a higher degree of uncertainty for IVW estimates" and none of the MR results for Menarche or Parity reached significance (as shown in Figure 2B). So it is unclear how can the authors justify their summary, Abstract "Older age of menarche was related to a more atherogenic metabolic profile in (MV and) MR". Albeit, the resolution of Figure 2B is too poor to see the p-value threshold used for MR but it seems different to that for MV - please clarify and justify this threshold and consider the overall conclusion.

Response to reviewer 1 (point 1):

We thank the reviewer for the comment. We have made substantial changes to the manuscript to facilitate visualisation of results. This encompasses changing Figure 2 to forest plots to facilitate comparison between multivariable and Mendelian randomization (IVW) results for non-derived metabolites. Measures that are derived or related to lipoprotein subfractions have been moved into supplementary text (Supplementary figures 1, 7 and 11). The description of results also takes this format and structure. Please see edits to main text and Figure 2 (panels A to C).

As expected, there is a higher degree of uncertainty in the IVW compared to multivariable regression estimates. To increase power in MR, we have included an additional set of IVW analyses using data for eight selected biomarkers assayed by clinical chemistry that match measures from the NMR metabolomics platform. These selected biomarkers are available in ~240,000 women (compared to ~65,000 women from our original analyses) and include albumin, ApoA1, ApoB, glucose, HDL-c, LDL-c, total cholesterol, and triglycerides. These results, now described in the text and presented in Supplementary figure 5, are largely in agreement with our main analyses and are estimated with high precision for age at menarche and menopause, as described in the text and pasted below:

Results – age at menarche (page 5, line 142):

"Following reviewer's comments, we repeated the IVW analyses for a larger sample of women (N=216,514-241,244) for the eight biomarkers assayed using clinical chemistry techniques that matched measures in the NMR metabolomics platform — i.e. albumin, apolipoprotein A1, apolipoprotein B, glucose, HDL-cholesterol, LDL-cholesterol, total cholesterol, and triglycerides. These results provided further evidence of older age at menarche being related to higher albumin, apolipoprotein A1, HDL-cholesterol, and lower triglycerides (P < 0.00093) (Supplementary Figure 5)."

Results – age at natural menopause (page 8, line 233):

“Repeating the IVW analyses in the larger sample of women (N=216,514-241,244) with selected biomarkers assayed by clinical chemistry confirmed that older age at natural menopause was related to lower albumin, LDL-cholesterol, and total cholesterol at $P < 0.00093$ (Supplementary Figure 5).”

Reviewer 1 (point 2):

Turning to Menopause, here they find several opposing effects for MV and MR on Cholesterol parameters, IDL, large medium and small LDL, and very small VLDL - these are illustrated together in Suppl Fig 6. Despite a number of sensitivity tests, the reason for this discrepancy is not identified. Opposite MR results are still found in within-siblings, Weighted median and MR Egger, and using main set SNPs. So MR and MV then are not so complementary approaches. Without knowing the reason for the difference it is unclear which direction is the correct one. One possible other difference between MV and MR is the selected population. MV is restricted to the 61% women who report reaching natural menopause. MR is in the all women. It is possible that the effects of menopause age differ somehow before and after menopause. This could be checked.

Response to reviewer 1 (point 2):

We have conducted follow-up analyses, which revealed that discrepant findings between multivariable regression and MR were explained due to sample selection in the first due to the exclusion of pre-menopausal women and a strong effect modification by chronological age in the association between age at natural menopause and metabolic traits, as detailed below:

In Results, we have added (page 8, line 237):

“We performed further analyses to investigate reasons underlying discrepant findings between multivariable and MR estimates for some metabolic measures. These analyses were restricted to the eight clinical chemistry biomarkers matching measures in the NMR platform to maximise statistical power since they have been measured in the full UK Biobank sample. First, we hypothesised that discrepant findings were related to differences in the sample used for multivariable regression, which excludes women with missing data on age at menopause (hereafter ‘selected sample’), and two-sample MR, which includes women even if they are missing data on age at natural menopause (hereafter ‘full sample’). To test that, we compared estimates from multivariable regression on the selected sample to MR on both the selected sample and full sample. In agreement with our hypothesis, multivariable regression and MR estimates for LDL-cholesterol and related traits (i.e. apolipoprotein B and total cholesterol) are comparable when restricting to the selected sample. In contrast, for albumin, discrepant results were related to differences between multivariable regression and MR rather than between selected and full sample (Supplementary Figure 15). Second, given women with missing data at age at menopause are typically pre-menopausal and younger, we explored age-stratified multivariable and MR estimates, which revealed a strong effect modification by chronological age on the association of age at menopause and LDL-c and related traits: in younger women (≤ 50 y), later age at menopause is related to lower LDL-cholesterol, while the opposite is true for older women (> 58 y) (Supplementary Figure 15). Age-related differences were also observed for other biomarkers, such as albumin. This age patterned results were largely similar when excluding women using statins at baseline or with a history of using hormone replacement therapy (HRT) (Supplementary Figure 16).”

In the Discussion, we have added (page 12, line 374):

“Findings from our multivariable regression analyses for age at natural menopause should be interpreted with caution given 40% of women were excluded from these analyses as they had not experienced a natural menopause and 7% of the women had experienced menopause less than two years before study recruitment (when blood samples for NMR metabolomics were collected). In follow-up analyses, we have shown that discrepancy in findings between multivariable and MR for

LDL-related traits were related to the exclusion of younger pre-menopausal women in multivariable regression. In addition, age-stratified analyses revealed that age at menopause is related to lower LDL-cholesterol in younger women but higher LDL-cholesterol in older women. We speculate that the negative association among younger women (≤ 50 y) reflects the fact that later menopause in this group will imply that women have only recently experienced menopause, and, therefore, have lower LDL-cholesterol compared to women with earlier menopause who had been postmenopausal for several years. Among older women (> 58 y), we found evidence of a positive association between age at menopause and LDL-cholesterol. We further explored these findings by excluding current statin and HRT users, which did not seem to substantially alter our findings, although such analyses should be interpreted with caution given the potential for collider stratification bias. To clarify the effect of age at menopause on all NMR metabolic measures using MR requires larger sample sizes than currently available.”

In the Supplementary results, we have added Supplementary figures 15 and 16, as shown below:

Suppl Fig 15. Age-combined and age-stratified estimates for the association between age at natural menarche and clinical chemistry biomarkers using multivariable regression (red) and one-sample Mendelian randomization (blue) restricted to women with data on age at menopause or two-sample Mendelian randomization (black) using data from all women.

Multivariable regression was performed for women with non-missing data on age at natural menopause using ordinary least squares (OLS) regression. One-sample Mendelian randomization was performed for women with non-missing data on age at natural menopause using two-stage least square (2SLS) regression. Two-sample Mendelian randomization was performed for all women regardless of missing data on age at natural menopause using the inverse variance weighted (IVW) method.

Suppl Fig 16A. Age-combined and age-stratified estimates for the association between age at natural menarche and clinical chemistry biomarkers excluding users of statins at baseline estimated using multivariable regression (red) and Mendelian randomization (blue) restricted to women with data on age at menopause or Mendelian randomization (black) using data from all women.

Multivariable regression was performed for women with non-missing data on age at natural menopause using ordinary least squares (OLS) regression. One-sample Mendelian randomization was performed for women with non-missing data on age at natural menopause using two-stage least square (2SLS) regression. Two-sample Mendelian randomization was performed for all women regardless of missing data on age at natural menopause using the inverse variance weighted (IVW) method.

Suppl Fig 16B. Age-combined and age-stratified estimates for the association between age at natural menarche and clinical chemistry biomarkers excluding users of hormone replacement therapy (HRT) at baseline estimated using multivariable regression (red) and Mendelian randomization (blue) restricted to women with data on age at menopause or Mendelian randomization (black) using data from all women.

Multivariable regression was performed for women with non-missing data on age at natural menopause using ordinary least squares (OLS) regression. One-sample Mendelian randomization was performed for women with non-missing data on age at natural menopause using two-stage least square (2SLS) regression. Two-sample Mendelian randomization was performed for all women regardless of missing data on age at natural menopause using the inverse variance weighted (IVW) method.

Reviewer 1 (point 3):

Other comments

P-value < 0.00076. This correction for 3 x 22 tests is based on a calculation of NMR metabolite data in a study of 14 cohorts in Ref 15. This threshold could easily be confirmed and calibrated to the current data. Suppl Fig 4, results for parity, the very wide X-axis scale means it is impossible to tell if MV results are significant. I suggest you use a tighter scale and truncate the longer confidence intervals for MR which are almost all non-significant.

Glycated Haemoglobin is mentioned in the results text but I can't find it in the Figures.

Figure 1. What is meant by "(no residual confounding)"? Surely this condition cannot be met.

Response to reviewer 1 (point 3):

As requested, we have performed principal component analyses in the NMR data used for this study, which results in 18 components explaining 95% of the variance. We have calibrated our multiple testing threshold p-value to reflect that, as described in the Methods section (page 17, line 562) and pasted below. Importantly, this did not alter the study's conclusions.

*"For both multivariable and MR analyses, we corrected for multiple testing using the Bonferroni method considering 3*18=54 independent tests (P=0.05/54≈0.00093). This was based on the three*

exposures included in our analyses (i.e. age at menarche, parity, and age at natural menopause) and the 18 independent features explaining over 95% of variance in the highly correlated NMR metabolic measures in our dataset as estimated by principal component analysis ⁵⁵”.

Following the reviewer’s comment, we have narrowed the scale of current Supplementary figure 7 (previous Supplementary figure 4) to improve visualisation of multivariable regression estimates.

Glycated haemoglobin was used in the within-sibling analyses given glucose was not available. Please see results in Supplementary figure 17. We have added further detail to the Methods section to emphasise that as follows: “... we used sex-combined data from a recent within-sibship GWAS, including up to 159,701 siblings from 17 cohorts, to test the effect of genetic susceptibility to higher age at menarche, parity and age at menopause on metabolic markers (i.e. LDL-cholesterol, triglycerides, HDL-cholesterol, C-reactive protein, and glycated haemoglobin)⁶². C-reactive protein and glycated haemoglobin were used as proxies for inflammation and hyperglycaemia, respectively, given GlycA and glucose were not available.”

In Figure 1, we have listed the key assumptions of each method as outlined in the figure’s title “*Overview of study design: key assumptions and biases*”. No residual confounding is listed under multivariable regression as this is a key assumption of the method.

Reviewer #2 (Remarks to the Author):

In this paper, using a triangulation approach, evidence is sought for the mechanisms relating female reproductive risk factors to cardiometabolic traits. This is a very large and very well conducted study finding that reproductive risk factors are associated with metabolic changes later in life. However, I think the conclusion may be a bit optimistic, given the fact that so very few associations seem to match with multivariable regression and MR.

This paper clearly contributes to the existing literature, even though most of the findings are not new, this is the first paper to take such an extensive approach. For all analyses, potential biases are taken clearly into account, and all MR assumptions are checked.

I have a few comments on this paper:

Reviewer 2 (point 1):

- My most important comment would be on the fact that this paper includes so many results, and it is often very hard to get a clear understanding from the tables and figures. So, it would be helpful to include some tables, that summarize the most important findings a bit more. For instance, a table with an overview for each determinant what traits were overlapping with the two (or three) methods with the estimates in there, so that I can quickly compare the results for those traits that in the end are considered to be causally related to the exposures.

Response to reviewer 2 (point 1):

We agree with the reviewer that the large amount of results presents a challenge to data visualisation. To address this, we have changed the presentation of the main results figure (i.e. Figure 2) and text. Please see details in our response to reviewer 1, point 1.

Reviewer 2 (point 2):

- The authors state that for the negative male analyses similar results in men and women would suggest that parity is not causally associated. I tend to disagree. It just means that it is not the pregnancy, but it could be due to just having kids. It is not being said that the SNPs identified for parity reflect pregnancy. In fact, one could question what kind of SNPs would come up with the number of children as an outcome, given that some of it is choice (many people choose to only have two children for instance), some of it is infertility of either the woman or the men involved, etc. which

may be similar in men and women. So what are we in fact measuring? Results show some differences between men and women, suggesting that there is some biological effect of the actual pregnancy. However, what keeps bothering me here is the additional effect of breast feeding, which is also associated with number of children, and the effect of that on BMI later in life. This in combination with the low power of this MR analysis, makes me wonder whether the authors might be a bit overoptimistic in their conclusions. Some of this needs to be incorporated in the discussion.

Response to reviewer 2 (point 2):

We agree with the reviewer that parity has a complex determination including an interplay between sociodemographic (e.g. education, age) and biological factors (e.g. fertility). However, we disagree with the reviewer's point that differences between female and male analyses do not answer our causal question.

In our study, we are interested in understanding the impact of women being exposed multiple times to the stress test of pregnancy and how this may impact their future metabolic health. This is challenging to study using conventional multivariable regression analyses given the complex and shared determinants of parity and future metabolic health and, therefore, the assumption of 'no residual confounding' in such analyses is unlikely to hold. For this reason, the use of men as negative controls is strategic since men are exposed to such risk factors that may influence their number of children and future health (e.g. education, aging, infertility, parenting) but cannot experience the stress test of a pregnancy. Therefore, we revised the text in the results section to give more emphasis to our motivation to use the negative control approach as follows (page 6, line 179):

"We used males as a negative control since men cannot experience the effects of being exposed to the stress test of pregnancy. Therefore, similar results between female and male analyses would be indicative of bias, such as by sociodemographic (e.g. education attainment) and biological (e.g. infertility) factors, rather than by an effect of pregnancy..."

We agree with the reviewer that the statistical power from our MR analyses for parity is limited, which is due to the low variance explained in parity by the genetic variants. We believe we have clearly acknowledged that throughout the text and hope future studies will increase power for such analyses once the full NMR metabolomics data is released, which will triple the current sample size.

We do not agree that factors downstream parity, such as breastfeeding or post-partum BMI, could bias our analyses as this would be on the pathway between parity and future metabolic health, and, therefore, mediators of any effect. As we were interested in the total (and not the direct) effect of parity, we do not believe it is appropriate to account for such factors in our analyses.

Reviewer 2 (point 3):

- Age at menarche: My main concern is that only linear associations were tested, while observational evidence suggests a non-linear association with cardiovascular disease. Might a non-linear analysis change the results?

Response to reviewer 2 (point 3):

We thank the reviewer for their suggestion and have now assessed non-linear relationships in sensitivity analyses. To assess whether there was a non-linear relationship between each reproductive trait and the non-derived metabolites, we compared the categorised reproductive trait entered into the model as a categorical variable and as a continuous variable using a likelihood ratio test. We plotted the results against the first reference category and reported the p-value for linear trend. Whilst there was some evidence of non-linearity for fatty acids and other lipids across all reproductive traits, overall, the assumption of linearity seemed to hold for most metabolites (34/55), see **Supplementary Table 6** and **Supplementary Figures 3, 9, and 13**. Furthermore, for those metabolites that did show evidence of non-linearity the confidence intervals overlapped. Of those that

showed evidence of non-linearity we fit restricted natural cubic spline models (with either 3, 4, or 5 knots placed at percentiles suggested by *Harrell*⁵⁸ for each reproductive trait). The models for *most* metabolites with the lowest AIC for age at menopause had 4 knots (**Supplementary Table 12**) and for age at menarche and parity had 3 knots (**Supplementary Tables 7 and 10**). For age at menopause and age at menarche this corresponded to ages 40, 49, 52, and 56, and ages 11, 13, and 15, respectively. For parity, knots were placed at 1, 2, and 3 births. We also compared these plots with plots from the main analysis model which assumes a linear relationship between the reproductive trait and each metabolite, see **Supplementary Figures 4, 10, and 14**. Whilst these plots showed some evidence of non-linearity, particularly for older ages of menarche and menopause where initial increases in lipids were observed and then started to plateau and in some cases decrease from ages ~13-15 and 55 onwards for age at menarche and age at menopause, respectively, most associations were linear or at monotonic across most of the distributions. We had less power and more uncertainty in the extremes of the reproductive traits distribution, for example for those with an age of menopause <40, suggesting a bigger sample size would be needed to determine evidence of non-linearity.

We have not included non-linear Mendelian randomization analysis given this requires very large sample sizes which are not currently available. There are plans for the NMR metabolite analyses to be completed on all UKB participants in the future, which, when completed, could provide power to address this.

We have added the following to the methods, results, and discussion:

Methods (page 18, line 579): *“In sensitivity analyses we assessed whether there was a non-linear relationship between each reproductive trait and 55 non-derived metabolites. For ease of presentation, we excluded measures that were derived (eg ratios) or related to lipoprotein subfractions as these are highly correlated with one or more of the 55 non-derived metabolites. We compared the categorised reproductive trait entered into the model as a categorical variable and as a continuous variable using a likelihood ratio test. Age at menarche and age at menopause were categorised into tertiles (<13 years, 13-14 years, >14 years) and quartiles (<49, 49-50, 51-53, >53), respectively. Parity was categorised as 0, 1, 2, and 3+. Results were plotted against the first reference category and the p-value for linear trend reported. For any metabolites that showed evidence of non-linearity, restricted cubic splines (with either 3, 4, or 5 knots placed at percentiles as suggested by Harrell⁵⁸ for each reproductive trait) were fit and compared to the main analysis model (assuming a linear association) using AIC (BIC and root mean square error also shown).”*

Results:

Overall (page 4, line 91)

“In sensitivity analyses, for the 55 non-derived metabolites, we categorised age at menarche, parity and age at natural menopause, tested for a linear trend and where there was evidence of non-linearity fit restricted cubic splines.”

Age at menarche (page 5, line 127)

*There was evidence of non-linearity between categories of age at menarche (<13, 13-14, >14 years) and 17 metabolites (**Supplementary Table 6 and Supplementary Figure 3**). Restricted cubic spline models (with 3 knots at ages 11, 13, and 15 years) generally showed an increase in albumin apolipoprotein A1, cholines, and phosphatidylcholines with older age at menarche until approximately age 13, in line with our linear association, and then began to flatten and/or decrease (**Supplementary Table 7 and Supplementary Figure 4**). Whilst older age at menarche was related to an increase in GlycA until ~13 years and then began to flatten.*

Parity (page 6, line 174)

There was some evidence of non-linearity between parity (0,1,2,3+) and 28 metabolites (Supplementary Table 6 and Supplementary Figure 9). However, restricted cubic spline models (with knots at 1, 2, and 3) generally showed monotonic relationships for those with no to four pregnancies, consistent with the main analysis models (Supplementary Table 10 and Supplementary Figure 10).

Age at menopause (page 7, line 212)

There was evidence of non-linearity across 24 metabolites (Supplementary Table 6 and Supplementary Figure 13) in the multivariable regression when menopause was categorised (<49, 49-50, 51-53, >53 years). Restricted cubic spline models (with 4 knots) were generally consistent with the main analysis (assuming a linear association) until age at menopause ~55 years when most levels of decreased (Supplementary Table 12 and Supplementary Figure 14).

Discussion (page 13, line 432): *“When assessing non-linearity, our multivariable regression results were generally consistent between the main analysis model (assuming a linear relationship) and categories for most metabolites. For metabolites that showed evidence of non-linearity, many seemed to plateau and decrease with older ages of menarche and menopause and similarly with higher parity (however, this was also where we had the least amount of data, which could be driving some of the non-linearity). We were unable to fit non-linear associations in an MR framework given this would require much larger sample sizes; future studies with larger sample sizes should be better powered to examine potential non-linear effects using MR and contrast those found in the multivariable regression.”*

Reviewer 2 (point 4):

Age at menopause: Results are very much in line with literature. I would suggest to include the paper of de Kat et al, as in that study we were really able to take age out of the association and found an effect of menopause on lipid levels, independent of age.

Response to reviewer 2 (point 4):

We thank the reviewer for the suggestion and have added the reference to the discussion: *“Previous conventional observational studies suggested older age at natural menopause to be associated with lower risk of cardiometabolic diseases risk³⁷ over and above the underlying age trajectory^{21,38}.*

Reviewer #3 (Remarks to the Author):

The authors analyzed the associations of age at menarche, menopause, and parity with 249 metabolic traits measured with nuclear magnetic resonance (NMR), utilizing the data from 65 487 UK Biobank women. The major strength of the paper, in addition to the large sample size, is leveraging evidence across 3 methods: multivariable regression analysis (MV), Mendelian Randomization (MR) and use of negative control (men) to reduce the confounding effects inherent in observational studies. Moreover, there's a paucity of well-conducted metabolomic studies in the reproductive field and the present study contributes important new evidence in this regard. The authors found that older age at menarche and menopause was associated with a more favorable metabolic profile in women and mixed results were obtained for parity. While the manuscript is overall very well written and the data and methodology are presented in a detailed way, I feel that some revisions are necessary to further strengthen the paper.

Reviewer 3 (point 1):

Abstract

1) The Abstract (p. 2) and Conclusions (p. 11) do not completely agree. While in the Abstract the authors claim that older age at menarche is related to a more proatherogenic profile (lines 28-29), in the Conclusions it is stated that “Overall, older age at menarche/menopause were [sic!] related to a more favorable metabolic profile” (lines 347-348). The authors should rephrase both parts and align the conclusions.

Response to reviewer 3 (point 1):

We thank the reviewer for spotting that and have corrected the text in the abstract accordingly as follows: “*Older age of menarche was related to a **less** atherogenic metabolic profile in MV and MR, which was largely attenuated when accounting for adult body mass index*”.

Reviewer 3 (point 2):

2) Overall, I recommend focusing on the results which were consistent/robust across MR and MV when drawing the conclusions.

Response to reviewer 3 (point 2):

We have focussed on agreement between MV and MR results and have changed Figure 2 to facilitate such comparison. Please see details in our response to reviewer 1, point 1.

Reviewer 3 (point 3):

Methods

3) Why weren't women who have never been pregnant used as a negative control? Since there is a plethora of sex-and gender-specific differences between women and men which might affect the metabolites and the associations with parity, including behavioral and biologic factors, involvement of inherent physiological differences, like the impact of the sex hormones, differences in cardiovascular risk factors, gender disparities from cultural and environmental factors, and differences between the sexes in the diagnosis and treatment of diseases, the authors might actually introduce new spurious confounding instead of controlling for it. The analyses with metabolites are cross-sectional, so the argument that some never-pregnant women might become pregnant in the future is less relevant.

Response to reviewer 3 (point 3):

We are unsure about the motivation for the reviewer's suggestion. In the parity analyses, never pregnant women are already included as the reference group. We have looked at the effect of parity (entered into the models as a continuous variable) on metabolites and therefore interpreted as the effect of parity on metabolites per 1 additional pregnancy. Also, we cannot see how never pregnant women could be used as a negative control for the analyses testing the association between parity and metabolic measures as there is no variability in their exposure to parity (i.e. parity = 0 for the never pregnant). Besides, as the reviewer points out, there may be differences in the confounding structure between male and female analyses, as we acknowledge in the paper, but the same is true for analyses comparing never to ever pregnant women.

Reviewer 3 (point 4):

4) Reported body size at age 10 (average, thinner and plumper) might not sufficiently capture the confounding nature of pre-exposure BMI, as these categories are imprecise, moreover, prone to recall bias. In fact, after considering the attained BMI at the time of metabolites measurement (post-exposure), most of the associations were attenuated (supplemental Figure 2) in MV and none survived the Bonferroni threshold in MR (Figure 2B) relating to parity or age at menarche.

Response to reviewer 3 (point 4):

We agree with the reviewer that reported body size at age 10 may not sufficiently capture BMI prior to the exposure. However, we do know that BMI in childhood is highly predictive of later BMI. Moreover, we cannot be confident that including covariates such as BMI, smoking and alcohol at the time of measurement (post-exposure) are not inducing bias that would attenuate associations to the null given they could be on the causal pathway (particularly age at menarche that occurs much earlier than BMI at the time of the assessment).

In the methods we have “By adjusting for the baseline variables at the first assessment we can either block the confounding path or create bias if these variables are mediators. If the results change between model (2) and (3) it is hard to distinguish whether its correct adjustment for confounding or whether it is a mediated path.”

Reviewer 3 (point 5):

5) Bonferroni correction: Could the authors please explain the rationale behind “This was based on the three exposures included in our analyses (i.e. age at natural menarche, parity, and age at natural menopause) and the 22 independent features estimated among the highly correlated NMR metabolic measures that explained over 95% of variance in these measures in an independent dataset using principal component analysis”, p. 14 lines 430-434. Since the authors chose an exploratory approach when examining the associations between metabolites and reproductive risk factors, I believe they should apply a more conservative method like correcting for all independent tests (all detected metabolites): $0.05/249 = 2.008 \times 10^{-4}$ or using an FDR correction based on all 249 tests.

Response to reviewer 3 (point 5):

Applying the more stringent correction suggested by the reviewer would be overly conservative. This is because many of the measures in the NMR data are highly correlated. Therefore, as done by previous studies⁵⁵, we have used principal components to identify the number of independent features explaining most variance in the data and used that to correct for multiple testing.

In addition, at the request of reviewer 1, we have re-run this analysis in our own dataset, as described in the response to reviewer 1, point 4.

Reviewer 3 (point 6):

6) Did the authors inspect the potential non-linearity between age at menarche/menopause and metabolites levels? This could be achieved with restricted cubic splines or fractional polynomials, for example. As a sensitivity analysis, I'd suggest a categorization of age at menarche and menopause.

Response to reviewer 3 (point 6):

We agree with the reviewer and have now added these results. Please see details in our response to reviewer 2, point 3.

Reviewer 3 (point 7):

Results

7) Table 1: why did the authors decide to present the characteristics stratified by sex, if the focus of the paper was female reproductive factors? Men are investigated only as negative controls for parity and are not the primary focus of the paper. It would be more sensible to present the data stratified by categories of (older) age at menarche, menopause, and parity (see also my comment regarding sensitivity analyses above).

Response to reviewer 3 (point 7):

We have now edited Table 1 to focus on females only and added categories of age at menarche, menopause and parity, and moved the characteristics of males to the supplementary (**Supplementary Table 2**). We have also stratified Table 1 by categories of each of the reproductive factors (**Supplementary Tables 1, 2, and 3**).

Reviewer 3 (point 8):

8) According to figure 2B (MR analyses), no metabolic measure was significantly associated with age at menarche or parity, and as seen from Figure 3, the MV and MR analyses do not agree with each other for the associations with parity and age at menopause. Moreover, the % of variability in reproductive factors explained by the selected SNPs was rather low (ranging from 0.2 to 8%, supplemental Table 2), and taking together, these data suggest that the results should be interpreted with caution.

Response to reviewer 3 (point 8):

Please see response to reviewer 1 - point 1, in which we explain in detail how we addressed substantial uncertainty in some MR results. Briefly, we have included an additional set of IVW analyses using data for eight selected biomarkers assayed by clinical chemistry that match measures from the NMR metabolomics platform. These selected biomarkers are available in ~240,000 women (compared to ~65,000 women from our original analyses) and include albumin, ApoA1, ApoB, glucose, HDL-c, LDL-c, total cholesterol, and triglycerides. These results, now described in the text and presented in Supplementary figure 3, are largely in agreement with our main analyses and are estimated with high precision for age at menarche and menopause.

Reviewer 3 (point 9):

Discussion

9) Regarding the associations of specific metabolites with parity (p. 9), a more cautious interpretation of potential causality is warranted, since a) the results between MV and MR are not in agreement, b) by using males as a negative control group the authors might introduce new confounding through the above-discussed mechanisms (see my comment #3)

Response to reviewer 3 (point 9):

We disagree that MV and MR results for parity are in disagreement. At the moment, the statistical power for this MR analyses is limited and, therefore, the higher degree of uncertainty limits inference on agreement between MV and MR.

Regarding the use of males as negative controls, please see our response to point 3. To highlight the confounding structure between females and males, we have added a table showing how potential confounders relate to parity and number of children (both summarised as 0, 1, 2, 3+) (**Supplementary Table 2**). We can see that potential confounders (age, BMI, education etc) are similarly distributed across categories of parity and number of children in males and females, respectively. This gives us more confidence that our assumption of 'similar confounding structure in males and females analyses' is plausible.

Reviewer 3 (point 10)

10) NMR has a relatively low sensitivity and, moreover, can detect lower number of metabolites as compared to LC-MS. The authors should address this limitation.

Response to reviewer 3 (point 10):

As requested, we have added to the discussion of limitations (page 15, line 451): *“In addition, the metabolic traits measured by the NMR metabolomics platform cover a limited set of metabolic pathways⁴⁶, and, therefore, future studies including data from more sensitive metabolomics techniques, such as mass spectrometry, may improve coverage of the metabolome and provide insights into additional biological processes related to reproductive events.”*

Reviewer 3 (point 11):

11) For the metabolites, robustly associated with reproductive risk factors across both the MV and MR analyses, can the authors maybe speculate about possible mechanistic pathways involved in those associations?

Response to reviewer 3 (point 11):

We believe we have already speculated about potential underlying mechanisms in the discussion section.

For age at menarche, we believe results are reflecting the close interplay between pubertal timing and adiposity; however, at present, it is challenging to tease apart whether adiposity is confounding or mediating a relation between pubertal timing and metabolic health, as highlighted in the discussion: *“There is evidence of a bi-directional relationship between puberty timing and adiposity, where pre-pubertal adiposity influences puberty timing, which in turn influences post-pubertal adiposity^{13,28,29}. In addition, genetic variants influencing age at menarche are known to influence BMI before and after puberty^{28,29}. The complex relationship between puberty timing and adiposity complicates inferences of the effect of age at menarche on the metabolic profile or disease risk in adulthood since the observed associations could reflect adult BMI mediating the effect of early age at menarche on metabolic measures or a confounding path from pre-puberty BMI.”*

For parity, we believe results potentially reflect repeated exposure to the physiological stress of pregnancy: *“Pregnant women undergo marked changes in physiology (e.g. lipid/glucose metabolism, adiposity, vascular function, hormone levels, and inflammatory response) and lifestyle (e.g. diet and physical activity³⁰) most of which return to their pre-pregnancy state after delivery^{20,30}. However, there are concerns that some of these changes might persist and accumulate over multiple pregnancies, impacting women’s cardiovascular health in the future, or that pregnancy acts as a stress test, unmasking an underlying high risk for cardiovascular disease^{30,31}.”*

For age at menopause, we have emphasized that *“The mechanisms underlying the putative protective effect of older menopause on the risk of metabolic diseases in MR studies is unclear, but might reflect an effect of prolonged exposure to sex hormones or of slower cell aging, given genetic variants associated with age at natural menopause are highly enriched for genes in DNA damage response pathways^{40,44}.”*

REVIEWER COMMENTS

Reviewer #1 (Remarks to the Author):

The paper has been significantly improved in response to my comments, particularly by extension of the MR analysis to include data from the much larger UK Biobank sample And also by performing stratified analysis that appear to explain the inconsistencies between MV and MR.

1. The directionally opposite associations between ANM and lipid profiles in younger versus older women is intriguing and should be mentioned in the abstract. How does this fit with previous reports and is there any plausible biological explanation? Furthermore it seems inappropriate to still conclude that older age at menopause is related to a more favourable metabolic profile (line 442).
2. The analysis using clinical chemistry data in the much expanded sample represent the only significant MR findings. Hence these results in supplementary figure 5 should be moved to the main paper.
3. Multivariable MR adjusting for BMI should also be performed in this larger sample
4. The figures have been revised as Forrest plots. describe what do the bars indicate ? are they 95% confidence intervals or corrected for multiple testing? if the former, there needs some further indicator for statistical significance. For several figures The resolution and wide X axis scale still make it hard to see whether MV results touch the null line.
5. Figure 1: it is unclear that the text in brackets are the key assumptions. I suggest to state "(Assumes...)"

Reviewer #3 (Remarks to the Author):

In my opinion, the authors did a great job in incorporating the referees' comments and suggestions and the manuscript improved substantially. However, I still do some major concerns as outlined below.

1) The authors should clarify if they are investigating the effect of pregnancy or parity. Based on the comment #2 for Reviewer #2, it seems that the authors are investigating the effect of being subjected to the stress of pregnancy, not parity. But again, in the discussion the authors make it clear that it is the exposure to parity (number of children) which is under focus in this study: "It is plausible that factors relating to metabolites, such as age, ethnicity, socioeconomic position, and BMI, relate similarly to number of children in females and males and hence that confounding structures are similar."

In their response to my concerns about using men as negative controls, the authors claim that never-pregnant women cannot be used as a negative control because there's no variability in their exposure to parity. However, negative control is the unexposed group by definition, that there would be no variability in the exposure is not surprising. If it is the effect of pregnancy and not parity they are investigating, the same argument applies to men (no variability in the exposure to pregnancy). Never-pregnant women would be a better negative control, since at least less bias would arise from gender-specific (social, environmental differences between men and women) and biological (sex hormones etc.) aspects, if both groups were women. If it is the effect of parity (i.e. having children = being fertile, and not pregnancy per se) the authors are examining, then men cannot be considered negative controls, because they are exposed (being fertile, having children). I strongly recommend to reconsider the definition of exposure (parity or pregnancy) and the group of negative controls.

2) Adjustments. I would agree with the authors that including such confounders as smoking, BMI, alcohol intake etc. would be adjusting for post-exposure mediators, if the authors would have measured the metabolites at the time of exposure (age at menarche, menopause etc). The major limitation of the current study is that the metabolites themselves are measured post-exposure, so without any adjustment for classic confounders it is not clear what associations the authors are actually capturing. Besides, the authors do adjust for education, which is technically attained in full after age at menarche onset (post-exposure).

3) Parity. I agree with other referees that claims about causality in this aspect are too optimistic, given my concerns about the selection of the negative group and that the MV findings are not backed up by MR analyses. Apparently even in larger sample size the MR analyses were not in line with the MV analyses, so is really imprecision and lack of power or rather no causal effect of parity (or pregnancy, however the authors define it) on metabolome in the present study?

4) Discussion. My comment in the previous revision round was about potential mechanisms/pathways with specific metabolites which the authors identified to be associated with age at menarche, menopause and parity. I do see that the authors speculate about biological context, however, the discussion is too generic without mentioning how the specific metabolites could be involved.

Reviewer #4 (Remarks to the Author):

The authors has done a great job responding to previous concerns raised by the reviewers. The presented work is very comprehensive and well-organized. This reviewer agree with previous reviewer on the need for summary boxes or figures that articulate take home messages for each evaluated characteristic in a simple manner.

In page 6, line 180, the authors mentioned figure 4. The mentioned figure is not available among uploaded figures relative to this manuscript.

REVIEWER COMMENTS

We thank the editor and reviewers for their helpful responses. We have responded below, with reviewers' comments in blue and our responses in black. Where we have added new text, we have this in italics (or highlighted) and then in the main text document tracked these changes.

Reviewer #1 (Remarks to the Author):

The paper has been significantly improved in response to my comments, particularly by extension of the MR analysis to include data from the much larger UK Biobank sample And also by performing stratified analysis that appear to explain the inconsistencies between MV and MR.

Reviewer 1 (point 1):

The directionally opposite associations between ANM and lipid profiles in younger versus older women is intriguing and should be mentioned in the abstract. How does this fit with previous reports and is there any plausible biological explanation? Furthermore it seems inappropriate to still conclude that older age at menopause is related to a more favourable metabolic profile (line 442).

Response to reviewer 1 (point 1):

We thank the reviewer for the comment.

We have made the following changes:

- a) Mentioned effect modification by age in the abstract:

"In MV and MR, older age at natural menopause was related to lower concentrations of inflammation markers, but inconsistent results were observed for LDL-related traits due to chronological age-specific effects."

- b) Clarified in results that age at natural menopause has a strong negative effect on LDL-related traits among younger women and a slightly positive effect among older women (lines 250-257):

*"Second, given women with missing data at age at menopause are typically pre-menopausal and younger, we explored age-stratified multivariable and MR estimates, which revealed a strong effect modification by chronological age on the association of age at menopause with LDL-c and related traits – e.g. older age at menopause is related to substantially lower LDL-cholesterol in younger women (≤ 50 y) (e.g. MV: -0.018 SD, 95%CI: $-0.021, -0.015$), but slightly higher LDL-cholesterol in older women (> 63 y) (e.g. MV: 0.004 SD, 95%CI: $0.003, 0.006$) (**Supplementary Figure 15**)."*

- c) We have expanded the discussion as follows (lines 386-403):

"In follow-up analyses, we have shown that discrepancy in findings between multivariable and MR for LDL-related traits were related to the exclusion of younger pre-menopausal women in multivariable regression. In addition, age-stratified analyses revealed that age at menopause is related to lower LDL-cholesterol in younger women but slightly higher LDL-cholesterol in older women. These results did not seem to be explained by higher intake of statins or HRT among older women, although such analyses should be interpreted with caution given the potential for collider stratification bias. Previous longitudinal studies indicated that LDL-cholesterol (1) and related traits increase sharply through the menopause transition and early postmenopausal years and then plateau with increasing postmenopausal years (2). In our cross-sectional analyses, we observed a non-linear pattern for several metabolites, such that mean metabolite levels increase linearly with age at menopause until 50-55 years old and then decline. Taken together, we speculate that these findings explain the pattern by chronological age in the association between timing of menopause and LDL-related traits. However, larger longitudinal studies with longer follow-up are needed to tease apart the complex nature, and possibly time-varying, effect of reproductive aging on the metabolome."

d) We have expanded the conclusion to reflect that the anti-atherogenic effects of slower reproductive aging was restricted to younger women:

“Overall, older age at menarche/menopause were related to a more favorable metabolic profile, while a mixed pattern was observed for higher parity. Evidence supporting a relation between later pubertal timing and a less atherogenic metabolic profile was largely explained by adult BMI, while findings supporting a relation between slower reproductive aging and a less atherogenic metabolic profile was mostly observed among younger women. These results could contribute to identifying novel markers for the prevention of adverse cardiometabolic outcomes in women and/or methods for accurate risk prediction.”

Reviewer 1 (point 2):

The analysis using clinical chemistry data in the much expanded sample represent the only significant MR findings. Hence these results in supplementary figure 5 should be moved to the main paper.

Response to reviewer 1 (point 2):

As requested, we have now added the figure to the main paper as Figure 3.

Reviewer 1 (point 3):

Multivariable MR adjusting for BMI should also be performed in this larger sample

Response to reviewer 1 (point 3):

As suggested by the reviewer, we have added multivariable MR results for age at menarche accounting for adult BMI in the larger sample of participants with clinical chemistry measures for the eight markers matching NMR measures in supplementary figure 6 (also pasted below). This addition did not change previous conclusions as detailed below:

*“Given the a priori evidence of bidirectional effects between age at menarche and BMI, we also performed multivariable IVW accounting for adult BMI to estimate the direct effects of age at menarche on metabolic measures, which resulted in estimates partly or completely attenuated to the null for most metabolic measures with few exceptions, such as glutamine and glycine (**Supplementary Figure 5 and 6**).”*

Suppl Fig 6. Univariable and multivariable Mendelian randomization estimates for the relation between older age at menarche and clinical chemistry biomarkers among females

Reviewer 1 (point 4):

The figures have been revised as Forrest plots. describe what do the bars indicate ? are they 95% confidence intervals or corrected for multiple testing? if the former, there needs some further indicator for statistical significance. For several figures The resolution and wide X axis scale still make it hard to see whether MV results touch the null line.

Response to reviewer 1 (point 4):

We thank the reviewer for the suggestion and have added this information to the footnote of the forest plots (Figures 2-4) as follows:

“Circles denote point estimates and indicate p-value < 0.00093 (closed circle) or ≥ 0.00093 (open circles). Horizontal bars denote 95% confidence intervals.”

We appreciate that the X axis scale for Figure 2B is wide, which reflects the high degree of uncertainty in the MR results for parity. We believe it is important that the figure reflects the uncertainty in these estimates. To help with visualising the results, the circles representing point estimates were filled only when estimates passed our multiple testing corrected p-value threshold.

Reviewer 1 (point 5):

Figure 1: it is unclear that the text in brackets are the key assumptions. I suggest to state "(Assumes...)"

Response to reviewer 1 (point 5):

We have now updated Figure 1 to include 'key assumption' inside the brackets.

Reviewer #3 (Remarks to the Author):

In my opinion, the authors did a great job in incorporating the referees' comments and suggestions and the manuscript improved substantially. However, I still do some major concerns as outlined below.

Reviewer 3 (point 1):

The authors should clarify if they are investigating the effect of pregnancy or parity. Based on the comment #2 for Reviewer #2, it seems that the authors are investigating the effect of being subjected to the stress of pregnancy, not parity. But again, in the discussion the authors make it clear that it is the exposure to parity (number of children) which is under focus in this study: "It is plausible that factors relating to metabolites, such as age, ethnicity, socioeconomic position, and BMI, relate similarly to number of children in females and males and hence that confounding structures are similar."

In their response to my concerns about using men as negative controls, the authors claim that never-pregnant women cannot be used as a negative control because there's no variability in their exposure to parity. However, negative control is the unexposed group by definition, that there would be no variability in the exposure is not surprising. If it is the effect of pregnancy and not parity they are investigating, the same argument applies to men (no variability in the exposure to pregnancy). Never-pregnant women would be a better negative control, since at least less bias would arise from gender-specific (social, environmental differences between men and women) and biological (sex hormones etc.) aspects, if both groups were women. If it is the effect of parity (i.e. having children = being fertile, and not pregnancy per se) the authors are examining, then men cannot be considered negative controls, because they are exposed (being fertile, having children). I strongly recommend to reconsider the definition of exposure (parity or pregnancy) and the group of negative controls.

Response to reviewer 3 (point 1):

We thank the reviewer for the comment and have edited the introduction and results section (please see highlights below) clarifying that we are interested in parity as a marker of repeated exposure to the physiological challenges of pregnancy as highlighted:

a) In the introduction (lines 63-74):

"The aim of this paper is to explore the extent to which women's reproductive markers have a causal effect on 249 metabolic measures (covering lipids, fatty acids, amino acids, glycolysis, ketone bodies and an inflammatory marker). We focus on three reproductive traits that represent key events in women's reproductive lives: (i) age at menarche, a marker of puberty timing, (ii) parity, a marker of repeated exposure to the physiological challenges of pregnancy, and (iii) age at menopause, a marker of reproductive aging. We explore the causal relationships between reproductive markers and metabolic measures..."

b) In results (lines 97-106 and 181-185):

"For the second approach ('negative control design' – only applicable for parity), we used linear regression models to test whether number of children was associated with metabolic measures among men; given men do not experience the repeated physiological stress of pregnancy, but are likely to demonstrate the same associations of confounders (eg. socioeconomic position, BMI, smoking) with number of live births, similar associations of number of live births with metabolic measures between men and women would indicate bias (eg due to confounding) rather than a causal effect of being exposed to the physiological stress of pregnancy on women's metabolic profile."

"Therefore, similar results between men and women would be indicative of bias, such as due to confounding by sociodemographic (e.g. education attainment) and biological (e.g. infertility) factors, rather than by an effect of repeated exposure to pregnancy."

c) In the discussion (lines 335-342):

"Pregnant women undergo marked changes in physiology (e.g. lipid/glucose metabolism, adiposity, vascular function, hormone levels, and inflammatory response) and lifestyle (e.g. diet and physical activity (3)), most of which return to their pre-pregnancy state after delivery (3, 4). However, there are

concerns that some of these changes might persist and accumulate over multiple pregnancies, impacting women's cardiovascular health in the future, or that pregnancy acts as a stress test, unmasking an underlying high risk for cardiovascular disease (3, 5). We used parity as a marker of being exposed to the physiological stress of multiple pregnancies."

We respectfully disagree with the reviewer's statement that '*negative control is the unexposed group by definition*' for two reasons:

- a) The unexposed group is not the negative control but the reference group. In our analyses, the unexposed (reference) group are women who have never experienced a live birth.
- b) The principle of a negative control exposure analyses relies on comparing the association between the exposure of interest and the outcome with the association between a control variable and the same outcome. This control variable ('negative exposure') is deliberately chosen on the assumption that it has no causal effect on the outcome but shares the same biases structures (e.g. confounding) as the main analyses. A paternal/partner negative control has been widely used in epidemiology (<https://doi.org/10.1093/ije/dyx213>) because it fulfils these criteria.
- c) Lastly, the suggestion that we use the no delivery of a live born as the negative control is statistically not possible. We could not fit a regression model in women or men with a variable for which everyone has the same value (i.e. here only those with 0 value for no live born children) with metabolomic traits.

Reviewer 3 (point 2):

Adjustments. I would agree with the authors that including such confounders as smoking, BMI, alcohol intake etc. would be adjusting for post-exposure mediators, if the authors would have measured the metabolites at the time of exposure (age at menarche, menopause etc). The major limitation of the current study is that the metabolites themselves are measured post-exposure, so without any adjustment for classic confounders it is not clear what associations the authors are actually capturing. Besides, the authors do adjust for education, which is technically attained in full after age at menarche onset (post-exposure).

Response to reviewer 3 (point 2):

We respectfully disagree with the reviewer's statement that 'The major limitation of the current study is that the metabolites themselves are measured post-exposure'; it is a strength, not a limitation, that metabolites are measured after the exposure. An established criterium for causal analyses is that the exposure is measured and/or occurred before the outcome is measured. Metabolites are the outcomes and, therefore, temporal relations imply they would come after the exposure. Given the definition of a confounder - that it is a known or plausible cause of the exposure and outcome - it is essential that confounders are measured or occur before the exposure. In our study, confounders, such as smoking, BMI, and alcohol intake are also taken at the same time as the outcome – at a mean age 56.4 years – these were measured after our exposures of interest – and in particular, a long time after age at menarche. For this reason, we chose our set of minimally adjusted confounders as age (although measured post exposure reproductive traits cannot cause (older) age), measure of body size at age 10 and education. Although, as the reviewer notes, for age at menarche, education will have occurred post-exposure, a person's educational attainment is influenced by parental education, income and occupation, and as such can be considered a measure of childhood as well as adult SEP (6). We therefore chose a priori to use education as a proxy for SEP. Furthermore, given that age at menarche is unlikely to influence educational attainment, we feel we are appropriately adjusting for confounding by SEP rather than adjusting for a mediator.

We have now added the following to the methods (line 592):

“For age at menarche, education will have been measured after the exposure. However, as it is influenced by parental education, income and occupation (occurring before menarche) and unlikely to be determined by age at menarche, we a priori considered this as a proxy of early life SEP.”

Reviewer 3 (point 3):

Parity. I agree with other referees that claims about causality in this aspect are too optimistic, given my concerns about the selection of the negative group and that the MV findings are not backed up by MR analyses. Apparently even in larger sample size the MR analyses were not in line with the MV analyses, so is really imprecision and lack of power or rather no causal effect of parity (or pregnancy, however the authors define it) on metabolome in the present study?

Response to reviewer 3 (point 3):

We agree that MR results for parity are very imprecise, as acknowledged in the paper, due the selected parity SNPs explaining little variability in the exposure (total $R^2 = 0.32$). This limits inferences regarding a causal effect of parity based on MR results. This is also consistent with the estimates not being precisely null which gives us more confidence that it is likely due to imprecision and a lack of power rather than there being no causal effect. On the other hand, results from multivariable analyses and negative controls were much more precisely estimated.

Reviewer 3 (point 4):

Discussion. My comment in the previous revision round was about potential mechanisms/pathways with specific metabolites which the authors identified to be associated with age at menarche, menopause and parity. I do see that the authors speculate about biological context, however, the discussion is too generic without mentioning how the specific metabolites could be involved.

Response to reviewer 3 (point 4):

We thank the reviewer for the suggestion and have extended the biological context and plausibility (please also see our response to reviewer 1 (point 1) with some examples for parity and age at menopause in relation to specific metabolites. However, we avoided discussing biological mechanisms/pathways for specific metabolites in detail since we have analysed 249 metabolic measures and the biological mechanisms underlying our findings are of interest but not part of the scope of the study.

In the discussion (lines 353-356 and lines 425-427):

A possible mechanism is that higher parity leads to greater insulin resistance in pregnant women and subsequently increases the production and secretion of hepatic triglycerides, which can lead to an increased lipid content in VLDL particles. (lines 353-356)

Moreover, the lipid metabolism is regulated by estrogen, meaning that lower levels of estrogen during menopause can cause an increase in lipids, particularly LDL, HDL, and triglycerides (lines 425-427)

Reviewer #4 (Remarks to the Author):

The authors has done a great job responding to previous concerns raised by the reviewers. The presented work is very comprehensive and well-organized.

Reviewer 4 (point 1):

This reviewer agree with previous reviewer on the need for summary boxes or figures that articulate take home messages for each evaluated characteristic in a simple manner.

Response to reviewer 4 (point 1):

We thank the reviewer and have added the following take home messages to a summary box and included some specific metabolites as examples:

Summary box**What is new?**

- Markers of women's reproductive health are associated with several common chronic conditions. Whilst some attempts have been made to explore the extent to which these associations are causal, metabolites could act as mediators of the relationship between reproductive markers and chronic diseases.
- Older age of menarche was related to a less atherogenic metabolic profile in multivariable regression and Mendelian randomization, however, this was largely attenuated when accounting for adult body mass index.
- In multivariable regression, higher parity related to complex changes in lipoprotein-related traits. Whilst these were not observed in male negative controls, suggesting a potential causal effect in females, they were not replicated in the Mendelian randomization, possibly due to imprecise estimates.
- Older age at natural menopause was related to lower concentrations of inflammation markers in both multivariable regression and Mendelian randomization. Consistent results were observed for LDL-related traits when stratified by chronological age.

Implications

- Given that the age at menarche results were largely attenuated to the null when accounting for adult BMI, it is likely that age at menarche itself may not causally relate to the metabolic profile.
- These results, particularly for parity and age at menopause, could contribute to identifying novel markers for the prevention of adverse cardiometabolic outcomes in women and/or methods for accurate risk prediction. For example, consistent with other studies, higher parity was associated with unfavourable (e.g. higher number of particles and lipid content in VLDL and higher glycine) changes in the metabolic profile. Similarly, older age at menopause was related to higher lipid content in HDL particles and lower systemic inflammation, as proxied by GlycA.

Reviewer 4 (point 2):

In page 6, line 180, the authors mentioned figure 4. The mentioned figure is not available among uploaded figures relative to this manuscript.

Response to reviewer 4 (point 2):

We thank the reviewer and have ensured that Figure 4 refers to the negative control analyses.

References

1. Clayton GL, Soares AG, Kilpi F, Fraser A, Welsh P, Sattar N, et al. Cardiovascular health in the menopause transition: a longitudinal study of up to 3892 women with up to four repeated measures of risk factors. *BMC Medicine*. 2022;20(1):299.
2. Matthews KA, Crawford SL, Chae CU, Everson-Rose SA, Sowers MF, Sternfeld B, et al. Are Changes in Cardiovascular Disease Risk Factors in Midlife Women Due to Chronological Aging or to the Menopausal Transition? *Journal of the American College of Cardiology*. 2009;54(25):2366-73.
3. Rich-Edwards JW, Fraser A, Lawlor DA, Catov JM. Pregnancy Characteristics and Women's Future Cardiovascular Health: An Underused Opportunity to Improve Women's Health? *Epidemiologic Reviews*. 2014;36(1):57-70.
4. Wang Q, Würtz P, Auro K, Mäkinen V-P, Kangas AJ, Soininen P, et al. Metabolic profiling of pregnancy: cross-sectional and longitudinal evidence. *BMC Medicine*. 2016;14(1):205.
5. Sattar N, Greer IA. Pregnancy complications and maternal cardiovascular risk: opportunities for intervention and screening? *BMJ (Clinical research ed)*. 2002;325(7356):157-60.
6. Bruna G, Mary S, Debbie AL, John WL, George Davey S. Indicators of socioeconomic position (part 1). *Journal of Epidemiology and Community Health*. 2006;60(1):7.

REVIEWERS' COMMENTS

Reviewer #1 (Remarks to the Author):

Apologies for revisiting my previous first comment, but this is a very complicated finding to digest and even harder to understand why.

1. The directionally opposite associations between ANM and lipid profiles in younger versus older women is intriguing How does this fit with previous reports and is there any plausible biological explanation?

The authors replied by citing some previous studies (lines 375 to 381) but these do not mention any directionally opposite associations with age. Auro (Ref 39) found that menopause was associated with higher lipid content in HDL particles. This fits the current finding but only in older women. Ref 41 is added which finds that HDL and non-HDL rise during the menopausal transition "with a stronger effect of chronological age". It is unclear how this sheds any light on the current findings.

Instead, the authors should acknowledge that these directionally opposite associations with age have not been previously noted.

For the plausible biological explanation, the authors have added text on lines 390 to 400 stating it is a) not due to statin or HRT use, b) possibly due to collider stratification bias and c) LDL increased sharply through the menopause transition and then plateaus.

For explanation b) can collider bias really explain non-linear associations between measured lipids and age at menopause shown in Suppl Fig 14? This does not seem plausible.

For explanation c) as stated above is unclear how these observed longitudinal patterns sheds any light on the current findings.

Line 399 appears to summarise these arguments but instead talks about implications of the directionally opposite associations with age rather than its explanation. "Taken together, we speculate that these findings explain the pattern by chronological age in the association between timing of menopause and LDL-related traits".

Reviewer #3 (Remarks to the Author):

I have no further concerns or comments.

REVIEWER COMMENTS

We thank the editor and reviewers for their helpful responses. We have responded below, with reviewers' comments in blue and our responses in black. Where we have added new text, we have this in italics (or highlighted) and then in the main text document tracked these changes.

Reviewer #1 (Remarks to the Author):

Apologies for revisiting my previous first comment, but this is a very complicated finding to digest and even harder to understand why.

1. The directionally opposite associations between ANM and lipid profiles in younger versus older women is intriguing and should be mentioned in the abstract. How does this fit with previous reports and is there any plausible biological explanation?

The authors replied by citing some previous studies (lines 375 to 381) but these do not mention any directionally opposite associations with age. Auro (Ref 39) found that menopause was associated with higher lipid content in HDL particles. This fits the current finding but only in older women. Ref 41 is added which finds that HDL and non-HDL rise during the menopausal transition "with a stronger effect of chronological age". It is unclear how this sheds any light on the current findings.

Instead, the authors should acknowledge that these directionally opposite associations with age have not been previously noted.

For the plausible biological explanation, the authors have added text on lines 390 to 400 stating it is a) not due to statin or HRT use, b) possibly due to collider stratification bias and c) LDL increased sharply through the menopause transition and then plateaus.

For explanation b) can collider bias really explain non-linear associations between measured lipids and age at menopause shown in Suppl Fig 14? This does not seem plausible.

For explanation c) as stated above is unclear how these observed longitudinal patterns sheds any light on the current findings.

Line 399 appears to summarise these arguments but instead talks about implications of the directionally opposite associations with age rather than its explanation. "Taken together, we speculate that these findings explain the pattern by chronological age in the association between timing of menopause and LDL-related traits".

Response to reviewer 1 (point 1):

We thank the reviewer and acknowledge this wasn't clear and have now rewritten the Age at menopause discussion section (line 378):

"Age at natural menopause

Observational studies suggest that menopause is associated with a worse cardiometabolic profile over and above chronological aging.(1-6) Previous cross-sectional(4)

and longitudinal studies(2, 5, 7) indicate that the menopause transition is associated with a shift towards a more atherogenic lipoprotein profile, such as characterized by higher concentration of apolipoprotein B and LDL-cholesterol, and possibly higher circulating glucose(5) and inflammatory markers(2, 4, 5). In addition, females experience a marked change in their metabolic profile at the age of late 40s and early 50s, which is not observed in males(4), providing further support for a role of menopause.

In our study, we focused on age at natural menopause as an indicator of reproductive aging. Findings from multivariable regression and MR were supportive of older age at menopause being related to lower systemic inflammation, as indicated by GlycA. On the other hand, MR indicated that older age at menopause is related to lower glucose and a less atherogenic lipoprotein profile (e.g. lower circulating apolipoprotein B and LDL-cholesterol) in line with previous studies, while multivariable regression did not support that. Multivariable regression results should be interpreted with caution as it was not possible to include ~40% of women, who did not have data on age at natural menopause, mostly due to being premenopausal (25%) or having a surgical menopause (12%). In addition, multivariable regression estimates were attenuated when excluding women reporting statins intake at baseline. In sensitivity analyses, we observed that multivariable regression and MR estimates are fairly consistent when stratified by chronological age, with older age at menopause being related to lower LDL-cholesterol in younger women (≤ 50 y) but slightly higher LDL-cholesterol in older women (> 58 y).

Previous longitudinal studies indicated that LDL-cholesterol(5) and related traits increase sharply through the menopause transition and early postmenopausal years and then plateau with increasing postmenopausal years (8) This is in line with our cross-sectional analyses, in which we observed a non-linear pattern for several metabolites, such that mean metabolite levels increase linearly with age at menopause until 50-55 years old and then decline. Taken together, these findings might explain the pattern by chronological age in the association between timing of menopause and LDL-related traits. Menopause is a continuous dynamic process of progressive decline in ovarian function and circulating estrogen levels. Therefore, we speculate that, among younger women (≤ 50 y) at baseline, those reporting a younger age at menopause are more likely to have fully experienced the menopause transition at the study baseline compared to those reporting an older age at menopause who may still be perimenopausal, which could explain the association between older age at menopause and lower LDL-related traits in this younger age group. On the other hand, among older than 58 years at baseline, most will have experienced the full menopausal transition; in this age group, women reporting younger age at menopause will have been postmenopausal for many years, while women reporting older at menopause might still be in their early postmenopausal

years, which might explain the association between older age at menopause and slightly higher LDL-related traits in this age group. In addition, we hypothesized these results could be related to higher intake of medications among older women, however, the chronological age-patterned results did not change substantially when excluding women reporting using statins or HRT at baseline. We note that such findings should be interpreted with caution given the lack of granularity in how we defined statins and HRT treatment and the potential for collider stratification bias (9). Although these results are intriguing, larger longitudinal studies with longer follow-up will be needed to tease apart the complex nature, and possible time-varying, effect of reproductive aging on metabolic profiles.

The largest two-sample MR analysis to date indicate that older age at menopause is related to lower risk of type 2 diabetes in females, but no difference in risk of cardiovascular disease or dyslipidemia (data combining males and females) (10). This is in agreement with our MR analyses suggesting older age at natural menopause is related to lower glucose, and with evidence from randomized controlled trials of estrogen therapy pointing to a protective effect on type 2 diabetes but no change in risk of cardiovascular diseases (11-13). The mechanisms underlying the putative protective effect of older menopause on the risk of metabolic diseases in MR studies is unclear, but might reflect an effect of exposure to sex hormones or of slower cell aging, given genetic variants associated with age at natural menopause are highly enriched for genes in DNA damage response pathways (10, 14). The consistent results between MR of age at menopause and randomized controlled trials of estrogen therapy for type 2 diabetes indicates that prolonged exposure to sex hormones is likely to be involved. Moreover, estrogen regulates LDL particle receptor and clearance from the circulation (15), meaning that the sharp decrease in levels of estrogen during menopause could plausibly explain some differences observed between menopause and lipoprotein traits (6).”

We have also now expanded the abstract as follows:

“In multivariable regression and Mendelian randomization, older age at natural menopause is related to lower concentrations of inflammation markers, but we observe inconsistent results for LDL-related traits due to chronological age-specific effects. For example, older age at menopause is related to lower LDL-cholesterol in younger women but slightly higher in older women.”

Reviewer #3 (Remarks to the Author):

I have no further concerns or comments.

Response to reviewer 3 (point 1):

We thank the reviewer for all their helpful comments previously.

References

1. de Kat AC, Dam V, Onland-Moret NC, Eijkemans MJC, Broekmans FJM, van der Schouw YT. Unraveling the associations of age and menopause with cardiovascular risk factors in a large population-based study. *BMC Medicine*. 2017;15(1):2.
2. Wang Q, Ferreira DLS, Nelson SM, Sattar N, Ala-Korpela M, Lawlor DA. Metabolic characterization of menopause: cross-sectional and longitudinal evidence. *BMC medicine*. 2018;16(1):17-.
3. Okoth K, Chandan JS, Marshall T, Thangaratinam S, Thomas GN, Nirantharakumar K, et al. Association between the reproductive health of young women and cardiovascular disease in later life: umbrella review. *BMJ*. 2020;371:m3502.
4. Auro K, Joensuu A, Fischer K, Kettunen J, Salo P, Mattsson H, et al. A metabolic view on menopause and ageing. *Nature Communications*. 2014;5(1):4708.
5. Clayton GL, Soares AG, Kilpi F, Fraser A, Welsh P, Sattar N, et al. Cardiovascular health in the menopause transition: a longitudinal study of up to 3892 women with up to four repeated measures of risk factors. *BMC Medicine*. 2022;20(1):299.
6. Karppinen JE, Törmäkangas T, Kujala UM, Sipilä S, Laukkanen J, Aukee P, et al. Menopause modulates the circulating metabolome: evidence from a prospective cohort study *European Journal of Preventive Cardiology*. 2022;29(10):1448-59.
7. El Khoudary SR, Chen X, Nasr A, Billheimer J, Brooks MM, McConnell D, et al. HDL (High-Density Lipoprotein) Subclasses, Lipid Content, and Function Trajectories Across the Menopause Transition. *Arteriosclerosis, Thrombosis, and Vascular Biology*. 2021;41(2):951-61.
8. Matthews KA, Crawford SL, Chae CU, Everson-Rose SA, Sowers MF, Sternfeld B, et al. Are Changes in Cardiovascular Disease Risk Factors in Midlife Women Due to Chronological Aging or to the Menopausal Transition? *Journal of the American College of Cardiology*. 2009;54(25):2366-73.
9. Pearce N, Lawlor DA. Causal inference—so much more than statistics. *International Journal of Epidemiology*. 2016;45(6):1895-903.
10. Ruth KS, Day FR, Hussain J, Martínez-Marchal A, Aiken CE, Azad A, et al. Genetic insights into biological mechanisms governing human ovarian ageing. *Nature*. 2021;596(7872):393-7.
11. Salpeter SR, Walsh JME, Ormiston TM, Greyber E, Buckley NS, Salpeter EE. Meta-analysis: effect of hormone-replacement therapy on components of the metabolic syndrome in postmenopausal women. *Diabetes, Obesity and Metabolism*. 2006;8(5):538-54.
12. Manson JE, Chlebowski RT, Stefanick ML, Aragaki AK, Rossouw JE, Prentice RL, et al. Menopausal Hormone Therapy and Health Outcomes During the Intervention and Extended Poststopping Phases of the Women’s Health Initiative Randomized Trials. *JAMA*. 2013;310(13):1353-68.
13. Gartlehner G, Patel SV, Viswanathan M, Feltner C, Weber RP, Lee R, et al. Hormone Therapy for the Primary Prevention of Chronic Conditions in Postmenopausal Women: An Evidence Review for the U.S. Preventive Services Task Force. Agency for Healthcare Research and Quality (US), Rockville (MD); 2017.
14. Day FR, Ruth KS, Thompson DJ, Lunetta KL, Pervjakova N, Chasman DI, et al. Large-scale genomic analyses link reproductive aging to hypothalamic signaling, breast cancer susceptibility and BRCA1-mediated DNA repair. *Nature Genetics*. 2015;47(11):1294-303.
15. Parini P, Angelin B, Rudling M. Importance of Estrogen Receptors in Hepatic LDL Receptor Regulation. *Arteriosclerosis, Thrombosis, and Vascular Biology*. 1997;17(9):1800-5.